# Ultrasound-activated piezo-hot carriers trigger tandem catalysis coordinating cuproptosis-like bacterial death against implant infections

Yanli Huang [1,8], Xufeng Wan[2,8], Qiang Su[3], Chunlin Zhao [4], Jian Cao[2], Yan Yue[2], Shuoyuan Li[2], Xiaoting Chen[5], Jie Yin [6], Yi Deng[7], Xianzeng Zhang [1,9] ✉, Tianmin Wu [1,9] ✉, Zongke Zhou [2,9] ✉ & Duan Wang [2,9] ✉

Implant-associated infections due to the formation of bacterial biofilms pose a serious threat in medical healthcare, which needs effective therapeutic methods. Here, we propose a multifunctional nanoreactor by spatiotemporal ultrasound-driven tandem catalysis to amplify the efficacy of sonodynamic and chemodynamic therapy. By combining piezoelectric barium titanate with polydopamine and copper, the ultrasound-activated piezo-hot carriers transfer easily to copper by polydopamine. It boosts reactive oxygen species production by piezoelectrics, and facilitates the interconversion between $Cu^{2+}$ and $Cu^+$ to promote hydroxyl radical generation via $Cu^+$-catalyzed chemodynamic reactions. Finally, the elevated reactive oxygen species cause bacterial membrane structure loosening and DNA damage. Transcriptomics and metabolomics analysis reveal that intracellular copper overload restricts the tricarboxylic acid cycle, promoting bacterial cuproptosis-like death. Therefore, the polyetherketoneketone scaffold engineered with the designed nanoreactor shows excellent antibacterial performance with ultrasound stimulation and promotes angiogenesis and osteogenesis on-demand in vivo.

Bone implantation utilizing inert substrates (e.g., metals, polymers, and ceramics) has emerged as a highly efficacious clinical stratagem in the realm of long bone fracture stabilization, spinal refurbishment, and replacement of arthritic joints[1,2]. Despite advancements in minimally invasive surgery and aseptic techniques, implant-associated infections (IAIs) continue to pose significant challenges in medical care and overall well-being. The existing clinical interventions are largely restricted to antibiotics and the physical removal of infected tissues or implants[3,4]. Unfortunately, the efficacy of systemically administered antibiotics is compromised by bacterial drug resistance and limited

[1]Key Laboratory of Opto-Electronic Science and Technology for Medicine of Ministry of Education, Fujian Provincial Key Laboratory of Photonics Technology, College of Photonic and Electronic Engineering, Fujian Normal University, Fuzhou 350117, China. [2]Orthopaedic Research Institute, Department of Orthopaedics, West China Hospital, Sichuan University, Chengdu 610041, China. [3]Department of Orthopedics, The Third Hospital of Mianyang, Sichuan Mental Health Center, Mianyang 621000, China. [4]College of Materials Science and Engineering, Fuzhou University, Fuzhou 350108, China. [5]Animal Experimental Center, West China Hospital, Sichuan University, Chengdu 610041, China. [6]Institute of Materials Research and Engineering, Agency for Science, Technology and Research, Singapore 138634, Singapore. [7]College of Biomedical Engineering, School of Chemical Engineering, Sichuan University, Chengdu 610065, China. [8]These authors contributed equally: Yanli Huang, Xufeng Wan. [9]These authors jointly supervised this work: Xianzeng Zhang, Tianmin Wu, Zongke Zhou, Duan Wang. ✉e-mail: xzzhang@fjnu.edu.cn; wtm@fjnu.edu.cn; zongke@126.com; wangduan_bone@163.com

drug penetration at the infection site. The development of smart coatings, which can either inherently identify and disrupt biofilm formation on implant surfaces or be activated remotely to provide antibacterial effects as needed, holds immense promise for effectively addressing IAIs[5–7].

Nanomaterials responsive to external stimuli with high tissue penetration and bio-safety, such as microwave and ultrasound (US), have been extensively studied for antibacterial treatments[8–10]. Piezoelectric nanomaterials, originating from the non-centrosymmetric crystalline architecture, have the innate capacity to engender an indigenous electric field and piezo-potential owing to the piezoelectric phenomenon, which induces catalytic behavior and generates reactive oxygen species (ROS) when subjected to US excitation[11]. Large piezo-potential and fast charge separation ability are two key factors for achieving effective sonodynamic therapy (SDT) in piezo-based sonosensitizer[12]. Therein, the hot carriers (electron-hole pairs) possessing higher energy above the thermal equilibrium, emerge as pivotal architects of surface redox dynamics. However, the mismatch between the ephemeral lifetimes of these hot carriers (on the order of femtoseconds to nanoseconds) and long timescales of chemical reactions (spanning milliseconds to seconds) constitutes a fundamental impediment that gives rise to feeble catalytic efficacy[13,14]. To surmount this challenge, researchers have conceived metal/semiconductor (Schottky) heterostructures, allowing hot electrons from metal surfaces with enough energy to transport across the Schottky barrier and be trapped at the conduction band (CB) of semiconductors[15,16], prolonging the lifetime of hot electrons and enhancing catalytic efficiency. Nevertheless, concerns regarding the potential biotoxicity of metals necessitate further exploration and investigation.

Combining SDT with chemodynamic therapy (CDT), which catalyzes the conversion of overexpression of $H_2O_2$ into hydroxyl radicals (•OH) via transition metal catalysts (TMCs)[17,18], may substantially boost ROS generation. Therein, Copper-based TMCs based on valence transition between $Cu^+$ and $Cu^{2+}$ have shown effective CDT effect. Besides, the anomalous copper accumulation in cells has been reported to directly bind to the tricarboxylic acid cycle (TCA) pathway with lipid-acylated components, consequently causing cuproptosis in tumor cells[19], providing new insights into copper-mediated bacterial death. For example, copper-doped polyoxometalate clusters can treat biofilm-associated infection by inducing bacterial cuproptosis-like death[20]. However, major limitations of current Cu-based TMCs include the easy oxidization and exhaustion of $Cu^+$. Moreover, some bacteria have developed various strategies to combat copper accumulation, including upregulating copper transport systems or producing copper-binding proteins that sequester excess Cu ions[21]. Therefore, developing Cu-based TMCs to yield explosive ROS via the synergy of SDT and CDT, changing membrane fluidity and permeability of bacteria, could facilitate bacterial copper overload and related metabolic interference. However, Cu ions have dose-dependent antibacterial performance and cytotoxicity[22], and whether Cu ions can induce cuproptosis-like bacterial death in a concentration lower than the minimal inhibitory concentration (MIC) and non-cytotoxic extracellular aggregation remains to be explored. To delve deeper into the intrinsic relationship between SDT and CDT synergistically enhancing ROS production, tandem catalysis, which enables rapid and selective synthesis of therapeutic molecules in live cells via TMCs-mediated chemical transformation[23–25], holds great potential to involve the sequential catalytic processes and achieve synergetic therapeutic effects. However, few previously reported Cu-based TMCs can be spatiotemporally triggered under external stimuli, especially ultrasound with high tissue penetration and safety for the human body.

Here, we proposed a multifunctional nanoreactor based on copper-armed piezoelectric sonosensitizer to achieve spatiotemporal US-driven piezo-hot carriers that assisted valence state interconversion between $Cu^{2+}$ and $Cu^+$. This approach enabled tandem catalysis between SDT and CDT, ultimately leading to efficient bacteria killing and addressing IAIs (Fig. 1). Specifically, a modified barium titanate (BT)-based (mBT) piezoelectric sonosensitizer with good SDT effect was encapsulated by polydopamine (pDA) with conjugated π structure serving as the "electron aspirator" to extract US-activated piezo-hot carriers outsides pDA (pBT) (Fig. 1b). The quantum transport characteristic of multilayer stacked pDA was elucidated using the Keldysh non-equilibrium Green's function formalism and density functional theory (NEGF-DFT). Subsequently, the electrons were transferred to $Cu^{2+}$ that chelated with pDA (CpBT), initiating the oxidizing reaction of $Cu^{2+}$ to $Cu^+$, which converted endogenous $H_2O_2$ into •OH. Subsequently, 3D-printed polyetherketoneketone (PEKK) bone implants, commonly utilized in orthopedics, were engineered with CpBT nanoreactors and hydroxyapatite (HA) to obtain the PH-CpBT scaffold. Notably, the PH-CpBT scaffold displayed excellent antibacterial performance against *Staphylococcus aureus* (*S. aureus*) with US stimulation by disturbing bacterial membrane homeostasis through elevated levels of ROS and promoting intracellular flux of $Cu^+$ into bacteria. Transcriptomics and metabolomics analysis revealed that intracellular Cu accumulation inhibited the TCA cycle by the combination of $Cu^+$ and lipoylated enzymes, eventually causing bacterial cuproptosis-like death. Moreover, the elevated levels of ROS generated by SDT and CDT of CpBT induced DNA and protein damage in bacteria. Besides, the PH-CpBT scaffold accelerated bone regeneration by promoting the angiogenesis by Cu ions and osteogenic differentiation of pre-osteoblasts by releasing Ca and P ions (Fig. 1c). We envisioned that the piezo-based nanoreactors triggering tandem catalysis to amplify therapeutic effect of SDT and CDT on bone implant surface can inhibit bacteria effectively, thus providing an on-demand and noninvasive way in response to the challenge of IAIs.

## Results and discussion
### Characterization of CpBT nanoreactors

The preparation process of CpBT nanoreactors was shown in Fig. 1a. The transmission electron microscopy (TEM) image of CpBT revealed a thickness of pDA of approximately 7 nm, and uniform distribution of Ti and Cu was displayed by the elemental mapping (Fig. 2a). The lattice fingers in high-resolution TEM image of CpBT (Fig. 2b) agreed well with the *d*-spacings of (101) planes of pure BT. The scanning electron microscopy (SEM) image showed CpBT possessed a uniform size distribution with an average size of several hundred nanometers (Supplementary Fig. 1a). X-ray diffraction (XRD) patterns of mBT, pBT and CpBT (Fig. 2c) exhibited pure perovskite structure, conforming to the standard pattern of pure BT. The Raman spectrum of mBT (Supplementary Fig. 1b) displayed a first-order Raman spectrum, with a peak at 305 cm$^{-1}$ suggesting the tetragonality. The average size of CpBT was around 210 nm and the zeta potential was −21 mV by the dynamic light scattering method, suggesting good stability in water (Supplementary Fig. 2). By the switching spectroscopy piezoresponse force microscopy (SS-PFM), the amplitude and phase images of mBT exhibited a standard butterfly amplitude curve and a near-complete phase shift of 180° at ±15 V, demonstrating the complete domain switching and strong intrinsic piezoelectric response (Fig. 2d). To assess the piezoelectricity of mBT and pure BT, the temperature dependence of the relative permittivity was measured (Fig. 2e). The relative permittivity of mBT at body temperature (37-43 °C) was much higher than pure BT, indicating the higher piezoelectricity of mBT. Additionally, the Landau free energy modeling revealed that at temperature of 37-43 °C, the polarization anisotropy energy and energy barrier (<0.3 J cm$^{-3}$) were relatively lower than that of pure BT (>1 J cm$^{-3}$) (Fig. 2f, Supplementary Figs. 3 and 4), which led to a small energy barrier for polarization rotation among T < 001> and O < 110> states[26], hence inducing the enhanced piezoelectric performance over the body temperature.

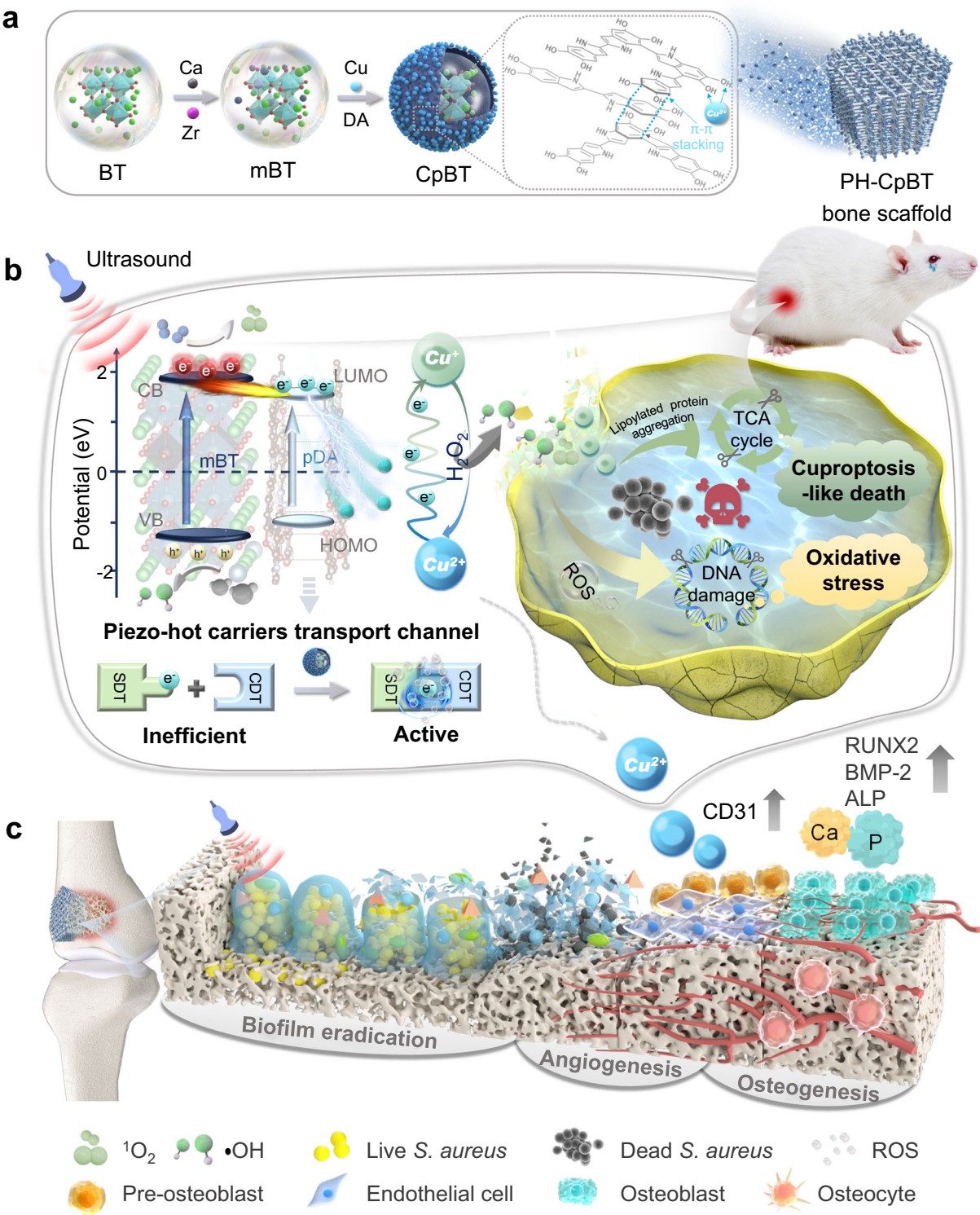

**Fig. 1 | Design strategy for US-induced tandem catalysis to trigger robust SDT and CDT. a** Fabrication of CpBT nanoreactors and PH-CpBT bone scaffold. **b** Driven by US-activated tandem catalysis, CpBT nanoreactors can trigger cascade reactions between SDT and CDT. *S. aureus* was eradicated through cuproptosis-like death and cell oxidative stress mediated by ROS. **c** In an infected bone implant model, the PH-CpBT bone scaffold showed good antibacterial, angiogenesis, and osteogenesis multifunctionality in vivo, synergistically.

## US-activated tandem catalysis of CpBT

To evaluate US-activated SDT and CDT performance, methylene blue (MB) was used to measure in vitro •OH generation. Over time and with increasing content of CpBT, CpBT gradually degraded MB under US stimulation (Supplementary Fig. 5a and b). Significantly, CpBT +$H_2O_2$ + US further enhanced the degradation of MB, while CpBT +$H_2O_2$ without US stimulation displayed poor degradation (Fig. 2g), highlighting the importance of US-mediated SDT and CDT in improving •OH generation. Additionally, we validated the production of singlet oxygen ($^1O_2$) using the fluorescent singlet oxygen sensor green (SOSG) probe, which showed a significant increase in $^1O_2$ production with CpBT under US stimulation (Supplementary Fig. 5c and 5d). To

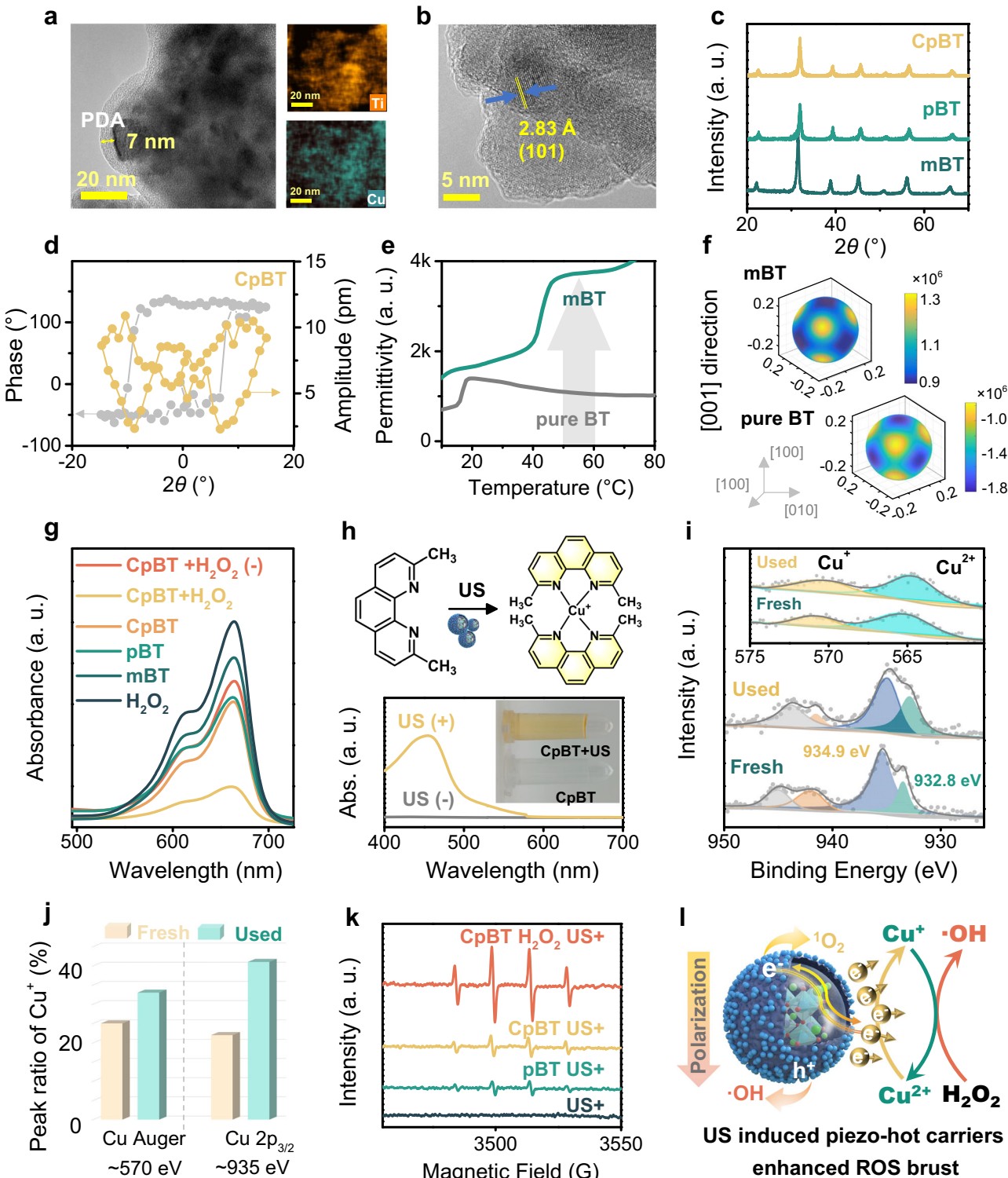

**Fig. 2 | Structural characterizations and US-activated tandem catalysis of CpBT nanoreactors. a** TEM image and elemental mapping of CpBT. Similar TEM images were obtained for more than three times experiments. **b.** HRTEM image of CpBT with lattice fringes corresponding to the (101) plane. Similar TEM images were obtained for more than three times experiments. **c** XRD patterns. **d** SS-PFM measurement. **e** The temperature dependence of relative permittivity of mBT and pure BT measured at 1 kHz. **f** Free-energy profiles for pure BT and mBT at 37 °C. **g** Comparison of MB degradation by mBT, pBT, CpBT, and CpBT+$H_2O_2$ under US stimulation. **h** The reaction mechanism of $Cu^+$ detection by neocuproine and the related UV-Vis absorbance spectra of neocuproine treated with CpBT and US stimulation. **i** XPS of Cu $2p_{3/2}$ for CpBT before and after US stimulation (the insets show Cu LMM spectra). **j** Peak ratio of $Cu^+$ to $Cu^{2+}$ at 570 eV and 932 eV in XPS spectra of CpBT before and after US stimulation. **k** •OH generation by ESR. **l** The proposed mechanism of enhanced ROS generation by US-activated tandem catalysis of SDT and CDT based on CpBT nanoreactors.

figure out the enhanced ROS production in the presence of $H_2O_2$, neocuproine, a $Cu^+$-specific sequestering agent, was used as an indicator. In the detection of $Cu^+$, colorless neocuproine typically formed yellow complexes $[Cu(neocuproine)_2]^+$ (Fig. 2h), showing maximum absorption at 452 nm. Neocuproine solution treated with CpBT+US turned yellow (Fig. 2h), providing strong evidence of the formation of $Cu^+$ on CpBT nanoreactor during US stimulation. Besides, the characteristic absorption peak increased slowly with US time increased, suggesting that the production of $Cu^+$ was time-dependent (Supplementary Fig. 6a). Without piezo-electrons produced by mBT, the reduction of $Cu^{2+}$ to $Cu^+$ under US stimulation failed (Supplementary Fig. 6b and 6c). Besides, XPS spectra of CpBT before and after US stimulation were carried out (Fig. 2i). The higher peak at ~935 eV in Cu $2p_{3/2}$ spectra was assigned to $Cu^{2+}$, accompanied by the characteristic $Cu^{2+}$ shakeup satellite peaks (938–945 eV). The lower peak at ~932 eV suggested the presence of $Cu^+$ or $Cu^0$ species. Furthermore, the Cu LMM Auger spectra at ~570 eV confirmed the presence of $Cu^+$ after US. Notably, the integral area ratio of $Cu^+$ to $Cu^{2+}$ after US was significantly enhanced at 935 eV (from 0.28:1 for fresh CpBT to 0.67:1 for used CpBT) and at 570 eV (from 0.33:1 for fresh CpBT to 0.5:1 for used CpBT) (Fig. 2j), indicating that part of surface $Cu^{2+}$ species were reduced to $Cu^+$ species during US stimulation. Furthermore, the electron spin resonance (ESR) technique was performed. The typical equal peaks with 1:2:2:1 represented •OH generation (Fig. 2k), and an apparent three-line spectrum with the peak intensity of 1:1:1 belonged to the signal of $^1O_2$ for both pBT and CpBT under US stimulation (Supplementary Fig. 5e). Therefore, US-excited $Cu^{2+}/Cu^+$ conversion of CpBT was proposed for tandem catalysis of SDT and CDT. Specifically, the electrons generated by sonosensitizer mBT under US stimulation were transported to $Cu^{2+}$ by "electron aspirator" pDA, which enabled the reduction of $Cu^{2+}$ to $Cu^+$ and transferred $H_2O_2$ to •OH (Fig. 2l) via Fenton-like reactions.

## Mechanism of US-activated tandem catalysis

The localized orbital locator (LOL) -π variant function was employed to investigate the delocalization channel of π electrons in pDA complicated system[27,28], and the isosurface maps of LOL-π for dopamine (DA), dihydroxyindole (DHI), and DHI oligomers were illustrated in Supplementary Fig. 7. The fully connected LOL-π isosurfaces of the six-membered rings in all DHI oligomers indicated a strong conjugation and the evident delocalization of π electrons in this region. Moreover, the π electron delocalization remained prominent in the layered aggregation system, with the LOL-π isosurfaces reflecting the extensive presence of delocalized π electrons over the six-membered rings (Supplementary Fig. 8). To elucidate carrier transportation between mBT and pDA, the NEGF-DFT was used[29]. The quantum transport architecture devices containing Au electrodes and 16-layer stacked DHI and DA acting as the scattering region were constructed (Supplementary Fig. 9a). The calculated potential energy distribution of the quantum transport architecture along the XY-plane was illustrated in Supplementary Fig. 9b. The transmission function, $T(E, V_b)$, were calculated using the *Landauer-Büttiker formula*[30]:

$$T(E, V_b) = Tr[\Gamma_L(E)G^R(E)\Gamma_R(E)G^A(E)] \tag{1}$$

where $G^R(E)$ and $G^A(E)$ represented the retarded and advanced Green's functions of the scattering region. $V_b$, $\Gamma_L(E)$, and $\Gamma_R(E)$ were the bias voltage, the linewidth functions of the left and right Au electrodes describing the coupling between electrodes and the scattering region, respectively. The calculated electronic structure showed that the constructed 16-layer stacked DHI and DA exhibited semiconducting properties (Figs. 3a and 3b). Due to its smaller electronic bandgap (2.61 eV) compared to mBT (3.16 eV, according to Fig. 3e), pDA served as a favorable charge transport medium during US stimulation, promoting the electron transfer from mBT to Cu ions. Notably, the

transmission spectra (Fig. 3c) demonstrated a zero transmission coefficient near the HOMO and LUMO, with no observable opened transport channel in the real-space scattering states below the LUMO energy level (<0.1 eV, 2 in Fig. 3d). The absence of transmission around the HOMO and LUMO energy levels concurred with the observations of a non-overlapping interlayer charge density in LOL-π isosurfaces within the layered aggregation structure. The carrier transmissions occurred beyond HOMO and LUMO, and the corresponding real-space scattering states of 16-layer stacked DHI and DA-based device at different energy levels clearly indicated the presence of available interlayer transport channels (Fig. 3d). According to valence band-XPS spectra (Fig. 3f), the valence band maximum (VBM) and the conduction band minimum (CBM) of mBT were calculated to be 1.90 eV and -1.26 eV, which was enough to trigger the redox reaction of -OH/•OH. Besides, the finite element analysis (Fig. 3g) demonstrated that the higher piezo-potential of mBT (0.035 V) compared with pure BT (0.010 V) (Supplementary Fig. 10) with the cavitation pressure of $10^8$ Pa, endowing full energy for US-induced hot carriers to transport. To assess the electron transmission in pDA-$Cu^{2+}/Cu^+$ composites, the charge density difference between pDA-$Cu^+$ and pDA-$Cu^{2+}$ complexes with different pDA configurations was calculated (Fig. 3h), showing that the oxidation of $Cu^+$ or the reduction of $Cu^{2+}$ led to significant charge transport in pDA-$Cu^{2+}/Cu^+$ composites. The negative charge density differences based on the natural population analysis of Cu ion [$Cu^+$-$Cu^{2+}$] further confirmed the reduction of charge on Cu ion in pDA-$Cu^+$ composites compared to the pDA-$Cu^{2+}$ composites. These findings provided strong evidence for charge transfer to Cu ion, consistent with the observations from the charge density difference isosurfaces. Thus, the involvement of pDA as a charge transport medium was essential for the charge transfer during the oxidation process of $Cu^+$ or the reduction process of $Cu^{2+}$. As a result, US-induced hot carriers by mBT with large piezo-potential injected into the enveloped pDA, which underwent interlayer carrier transport through high-energy scattering states, and were eventually transferred to $Cu^{2+}$ ions at the surface, enabling the reduction of $Cu^{2+}$, eventually displaying the intrinsic ability to assemble catalytic process for spatiotemporal control over cascade reactions like in living systems (Fig. 3i).

## Antibacterial activity of PH-CpBT scaffold in vitro

The 3D-printed PEKK bone scaffold (PEKK, P) was designed and modified with pDA (Pp) (Supplementary Fig. 11), and loaded with nano-HA (PH) and CpBT nanoreactors (PH-CpBT). Each rod within the PEKK scaffold had a diameter of approximately 300 μm, with an approximate gap of 200 μm between adjacent rods (Supplementary Fig. 12a). The CpBT was uniformly distributed on the scaffold (Supplementary Fig. 12b and 12c). The Young's modulus of PH-CpBT measured 248 Mpa, closely matching the range of human trabecular bone (6.9 to 199.5 Mpa) (Supplementary Fig. 13a and b). Furthermore, CpBT improved the hydrophilicity properties of the PEKK scaffold (as indicated by water contact angle, Supplementary Fig. 13c), facilitating the incipient adhesion of osteocytes. To assess the stability of CpBT coating on PH-CpBT with US stimulation, the released CpBT was detected by inductively coupled plasma (ICP) (Supplementary Fig. 14). Process A contained the released Cu ions and CpBT from scaffolds, and process B contained the released Cu ions only. It showed that the concentration of Cu of two processes by ICP was similar, indicating that CpBT on PH-CpBT surface was stable due to the strong adhesion of pDA[31].

Subsequently, in vitro antibacterial ability of different scaffolds against *S. aureus* and *Escherichia coli* (*E. coli*) was assessed (Supplementary Fig. 15). Under US stimulation, PH-CpBT with $H_2O_2$ addition exhibited an efficient antibacterial rate of 99.90% for *S. aureus* (Fig. 4a) and 99.95% for *E. coli* (Fig. 4b). In the absence of US, PH-CpBT displayed inefficient antibacterial properties for the unactivated SDT. Compared with Pp+US, PH + US, and PH-pBT, bacteria cultivated with

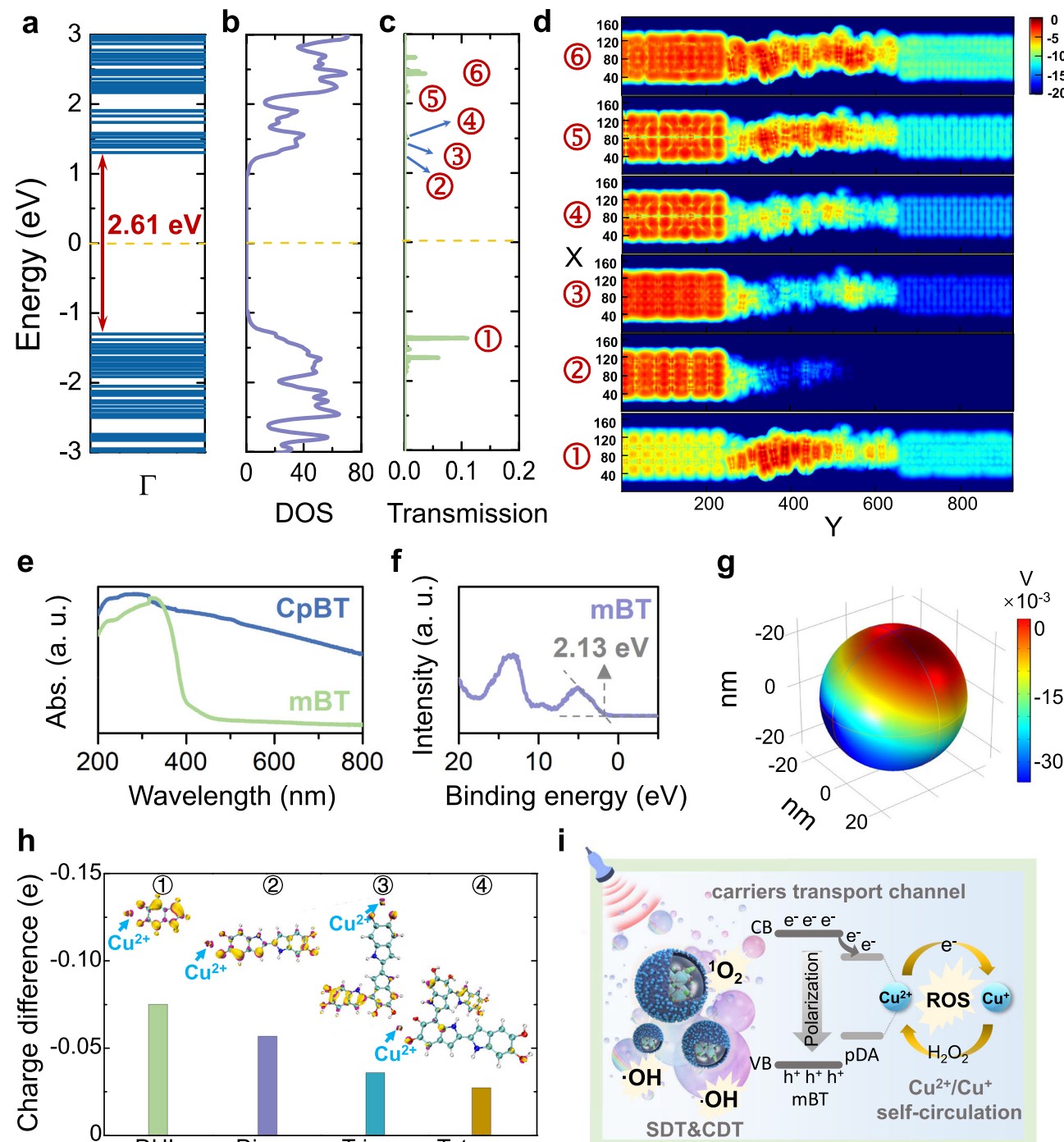

**Fig. 3 | Mechanism of US-activated tandem catalysis for enhancing SDT and CDT. a** Energy levels and **b** the corresponding density of states (DOS) of 16-layer stacked DHI and DA. **c** The transmission spectra of the 16-layer stacked DHI and DA-based transport architecture device. **d** The real space scattering states of the 16-layer stacked DHI and DA system at different energy levels. All the figures share the same color bar. **e** UV-Vis diffuse reflectance spectra of mBT. **f** VB-XPS spectra of mBT and CpBT. **g** Finite element method simulation for piezo-potential distribution on the surface of mBT. **h** The charge difference value achieved from natural population analysis of Cu ion (Cu$^+$-Cu$^{2+}$). Isosurface map of charge density difference computed from first principles for DHI-Cu$^+$-Cu$^2$, Dimer-Cu$^+$-Cu$^2$, Trimer-Cu$^+$-Cu$^2$, and Tetramer-Cu$^+$-Cu$^2$ (Isosurface in 0.003 $e$ Å$^{-3}$). Violet and yellow colors correspond to positive and negative differences, respectively. The silver gray sphere represented Cu ion. **i** The mechanism of US-activated tandem catalysis of SDT and CDT.

PH-pBT+US, PH-CpBT+US, and PH-CpBT+H$_2$O$_2$ photographed by SEM displayed varying degrees of deformation (Supplementary Fig. 16), while those in PH-CpBT+H$_2$O$_2$ + US exhibited wrinkled membranes, indicating the inactivation of bacteria. The bacterial biofilm was severely disrupted after PH-CpBT+H$_2$O$_2$ + US treatment, with wafery and dispersive features and intense red fluorescence observed, indicating that a mass of bacteria in biofilm were killed (Fig. 4c). To observe microscopic changes in dead bacteria, Bio-TEM images of bacteria were examined (Supplementary Fig. 17a). Bacteria incubated with PH-CpBT+H$_2$O$_2$ + US showed incomplete walls and cytoplasmic leakage in both *E. coli* (Fig. 4d) and *S. aureus* (Fig. 4e), and bacteria with H$_2$O$_2$ + US maintained intact morphology. Besides, the element mapping of bacteria showed intracellular copper content was distinct in PH-CpBT +H$_2$O$_2$ + US compared with US + H$_2$O$_2$ and PH-CpBT+H$_2$O$_2$

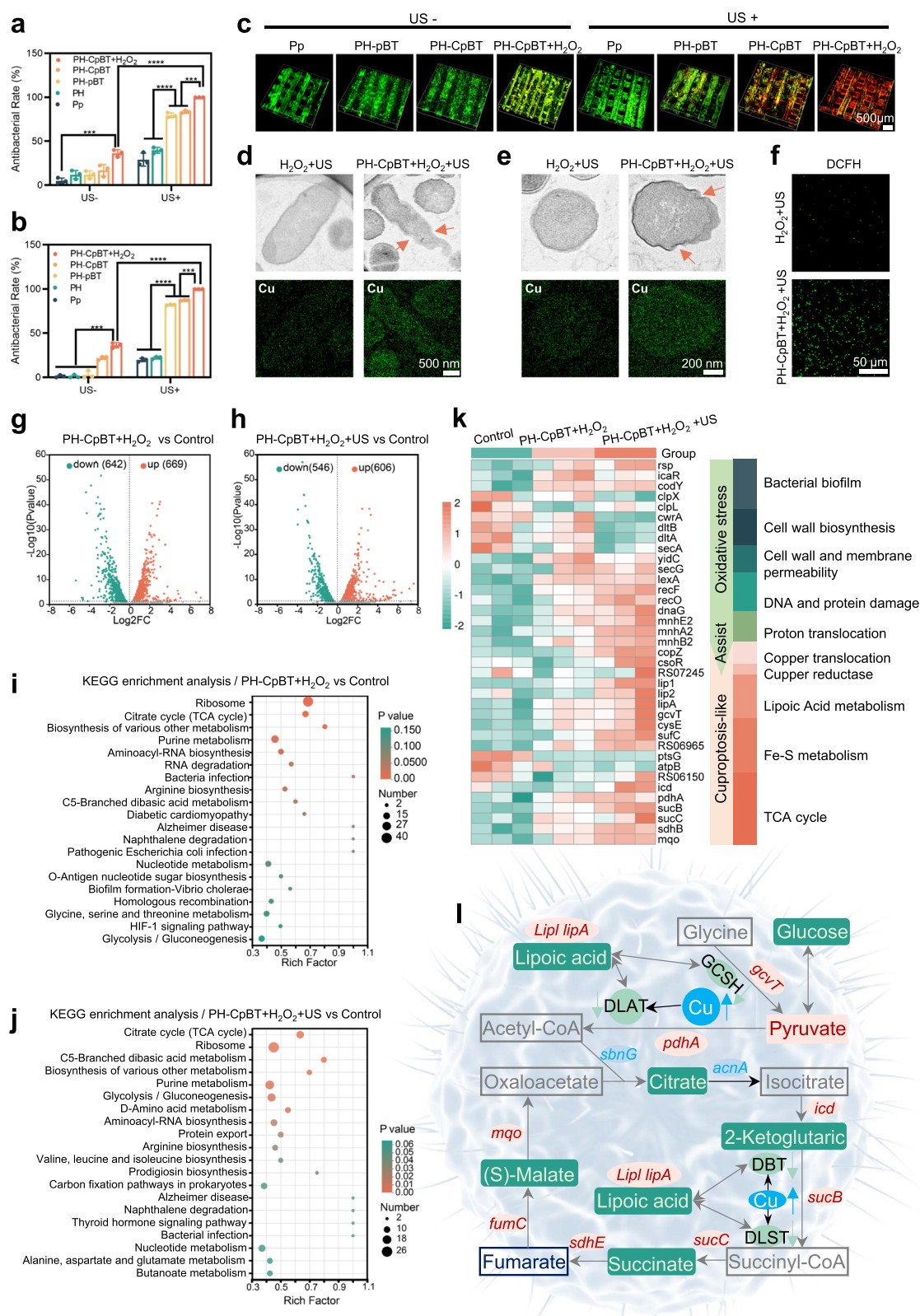

(Supplementary Fig. 17b), indicating an increased intracellular copper inward flow due to the membrane permeability alteration induced by evaluated ROS (Fig. 4f and Supplementary Fig. 18). Previous reports showed that Cu ions have dose-dependent antibacterial activity[22]. To find out the antibacterial effect of Cu ions in this work, Cu ions released from PH-CpBT scaffolds with or without US stimulation were analyzed

by an ICP spectrometer (Supplementary Fig. 19). The total amount of released Cu ions was 3-7 µg L⁻¹ for each PH-CpBT scaffold, much lower than MIC of Cu²⁺ ions (630 ug L⁻¹ for *S. aureus* and 63-630 ug L⁻¹ for *E. coli*)[32]. Furthermore, *S. aureus* or *E. coli*. were co-incubated with the leaching solution of PH-CpBT after US stimulation (30 min) for 24 h (Supplementary Fig. 20a), finding that the antibacterial rate was lower

**Fig. 4 | Antibacterial activity of PH-CpBT scaffold in vitro.** The antibacterial results of Pp, PH, PH-pBT, and PH-CpBT against **a** *S. aureus* and **b** *E. coli* with or without US stimulation. **c** 3D CLSM images of *S. aureus* biofilms by Live/Dead staining following incubation on different scaffolds with or without US stimulation. A representative image of three biological replicates from each group was shown. The microstructure and element mapping of Cu of **d** *S. aureus* and **e** *E. coli* observed by Bio-TEM. A representative image of three biological replicates from each group was shown. **f** The fluorescence image of intracellular ROS in *S. aureus* with DCFH probe. A representative image of three biological replicates from each group was shown. Volcano plot for the distribution of DEGs in **g** PH-CpBT+H$_2$O$_2$ and **h** PH-CpBT+US compared with control. KEGG enrichment of top 20 relevant pathways in response to **i** PH-CpBT+H$_2$O$_2$ and **j** PH-CpBT+ H$_2$O$_2$ + US. **k** Left, heat map showing

the differential expression of genes of interest. Right, the death pathways by ROS-induced oxidative stress and Cu-induced cuproptosis-like death. **l** A correlation analysis between differential metabolites and differentially expressed genes. Red rectangles indicated increased metabolites, and green rectangles indicate decreased metabolites; red circles indicated up-regulated genes, and green ovals indicated down-regulated genes. DBT: Dihydrolipoamide Branched Chain Transacylase E2, GCSH: Glycine Cleavage System Protein H, DLST: Dihydrolipoamide S-Succinyl transferase, DLAT: Dihydrolipoamide S-Acetyltransferase. **a, b** *n* = 3 biologically independent samples; ANOVA followed by Tukey's multiple comparisons; data were presented as mean values ± standard deviations (SD); error bars = SD. Significant differences between groups were indicated as ****$p < 0.0001$, ***$p < 0.001$, **$p < 0.01$, and *$p < 0.05$. Source data are provided as a Source Data file.

than 50%, indicating that the antibacterial performance by only released Cu ions was poor. Besides, bacteria were also incubated with PH-CpBT+US and tetrathiomolybdate (TTM, a copper chelator). In this situation, Cu ions cannot get into the bacteria, resulting in an antibacterial rate of 70% due to ROS attack (Supplementary Fig. 20b and 20e-g). Furthermore, mBT and Cu ions coating on Pp without pDA as interlayer exhibited a lower antibacterial rate than PH-CpBT (Supplementary Fig. 20c), suggesting that the electron transport within pDA was a key factor for the reduction of Cu$^{2+}$. These results proved that the antibacterial mechanism was more intricate than the release of Cu ions. In addition, we compared the antibacterial activity of CpBT with other piezoelectric materials (including BT, ZnO, MoS$_2$ and TiO$_2$), finding that CpBT showed better antibacterial activity (Supplementary Fig. 21). The leaking of bacterial proteins was detected using the protein leakage assay, and PH-CpBT exhibited the highest protein leakage, indicating significant membrane permeability alteration and cytoplasmic leakage after US-activated tandem catalysis (Supplementary Fig. 22a). Besides, the total bacterial DNA was quantified by Bacterial DNA Kit for different groups (Supplementary Fig. 22b). Notably, only a few intact DNA was detected in PH-CpBT+H$_2$O$_2$ + US, indicating severe damage to bacterial DNA due to the initiated ROS storm.

## Antibacterial mechanism of PH-CpBT scaffold

To gain insights into the comprehensive antibacterial effects of PH-CpBT on *S. aureus*, RNA-sequencing analysis was conducted. Volcano plots revealed that out of a total of 2622 expressed genes in both groups, and 642 genes were downregulated and 669 genes were upregulated in PH-CpBT+H$_2$O$_2$ without US (Fig. 4g) compared with Control. Furthermore, PH-CpBT+H$_2$O$_2$ + US exhibited 1152 significantly differentially expressed genes (DEGs), with 546 genes downregulated and 606 genes upregulated (Fig. 4h). Correlation analysis confirmed the comparability of three groups (Supplementary Fig. 23). The DEGs were conditioned to Kyoto Encyclopedia of Genes and Genomes (KEGG) annotation analysis, showing that both PH-CpBT+H$_2$O$_2$ and PH-CpBT+H$_2$O$_2$ + US had significant genes primarily enriched in membrane transport, folding and degradation, translation, lipid metabolism, and energy metabolism (Supplementary Fig. 24). KEGG pathway enrichment analysis and generated bubble plots depicting the top 20 metabolic pathways elucidated PH-CpBT+H$_2$O$_2$ had significant effects on *S. aureus* in ribosome, citrate cycle (TCA cycle), biosynthesis of various secondary metabolisms, purine metabolism, amonoacyl-tRNA biosynthesis, and RNA degradation (Fig. 4i). However, PH-CpBT +H$_2$O$_2$ + US predominantly affected the citrate cycle (TCA cycle), ribosome, C5-Branched dibasic acid metabolism, biosynthesis of various secondary metabolisms, purine metabolism, amonoacyl-tRNA biosynthesis, and protein export (Fig. 4j). Notably, TCA cycle pathway exhibited a high enrichment index and a small *p*-value in both enrichment analyses, which led us to delve into the concept of cuproptosis. Cuproptosis related to a type of cell death triggered by excessive copper that entered the cell and got reduced to the more toxic Cu$^+$ state. This process led to the aggregation of four enzymes linked to the lipoic acid in the TCA cycle, along with disruptions in Fe-S

cluster protein functionality[19]. Interestingly, upregulation or down-regulation of related genes was observed in PH-CpBT+H$_2$O$_2$ + US and PH-CpBT+H$_2$O$_2$, leading to a phenomenon named bacterial cuproptosis-like death[20]. These gene changes were more pronounced in PH-CpBT+H$_2$O$_2$ + US (Fig. 4k). This can be attributed to the eruptive ROS production in PH-CpBT+H$_2$O$_2$ + US facilitated by the synergistic SDT and CDT effect, which not only compromised the integrity of the bacterial cell wall and membrane but also affected the proton transport channel, leading to increased influx of copper ions into the bacterial cytoplasm (Fig. 4d and e). Consequently, copper overload transpired, resulting in changes in associated genes and a more pronounced cuproptosis-like death in PH-CpBT+H$_2$O$_2$ + US. An intriguing observation was the relatively inconspicuous upregulation of the RS07245 gene (a reductase, similar to *FDX1*) in PH-CpBT+H$_2$O$_2$ + US. This gene was responsible for the reduction of Cu$^{2+}$ to Cu$^+$, which was crucial to cuproptosis-like death process[33]. The limited upregulation of it in PH-CpBT+H$_2$O$_2$ + US may be attributed to the higher content of Cu$^+$ released by CpBT nanoreactors. It was known that Cu$^+$ exhibited stronger cuproptosis toxicity, which explained the greater observation of cuproptosis-like death in bacteria in PH-CpBT+H$_2$O$_2$ + US. In results, CpBT achieved the destruction of bacterial biofilm, induced DNA damage and protein leakage of bacteria through the toxicity of ROS-related oxidative stress, and the inhibition of TCA cycle by cuproptosis-like effects, ultimately culminating in complete sterilization. To present the changes in metabolites and the relevant genes in the TCA cycle more clearly, we conducted targeted detection of TCA cycle metabolites. We observed the lowest levels of metabolites in PH-CpBT+H$_2$O$_2$ + US compared with Control and PH-CpBT+H$_2$O$_2$ (Supplementary Fig. 25). Figure 4l vividly illustrated the changes in TCA cycle metabolites and the genes influenced by them. In short, under the influence of ROS, bacteria with PH-CpBT+H$_2$O$_2$ + US experienced an increase in intracellular Cu, which bound with four lipoylation enzymes (DLAT, GCSH, DBT, and DLST)[19,34], leading to their deactivation and consequent reduction in metabolites throughout the pathway. Additionally, due to the formation of more lipoylation proteins, lipoic acid was reduced[35], resulting in the upregulation of the genes that regulated lipoic acid synthesis[36]. Ultimately, the decreased metabolites, through negative feedback regulation, increased the expression of genes regulating the TCA cycle. Therefore, through metabolomics and transcriptomics, it was evident that copper-induced cuproptosis-like death, causing an overall reduction in metabolism, and in synergy with ROS, effectively killed bacteria.

## In vivo antibacterial capability

To assess the antibacterial performance in a live setting, *S. aureus*-contaminated scaffolds were implanted into femoral condyle bone defects in the experimental *Sprague Dawley* (SD) rats (Fig. 5a and Supplementary Fig. 26). The baseline weights of all groups were equivalent before surgery (Supplementary Fig. 27). Detection of IAIs in clinical scenarios typically relied on routine blood tests conducted on the second day after surgery. Therefore, we performed US stimulation at the implant site from the first to 6th-day post-surgery to replicate

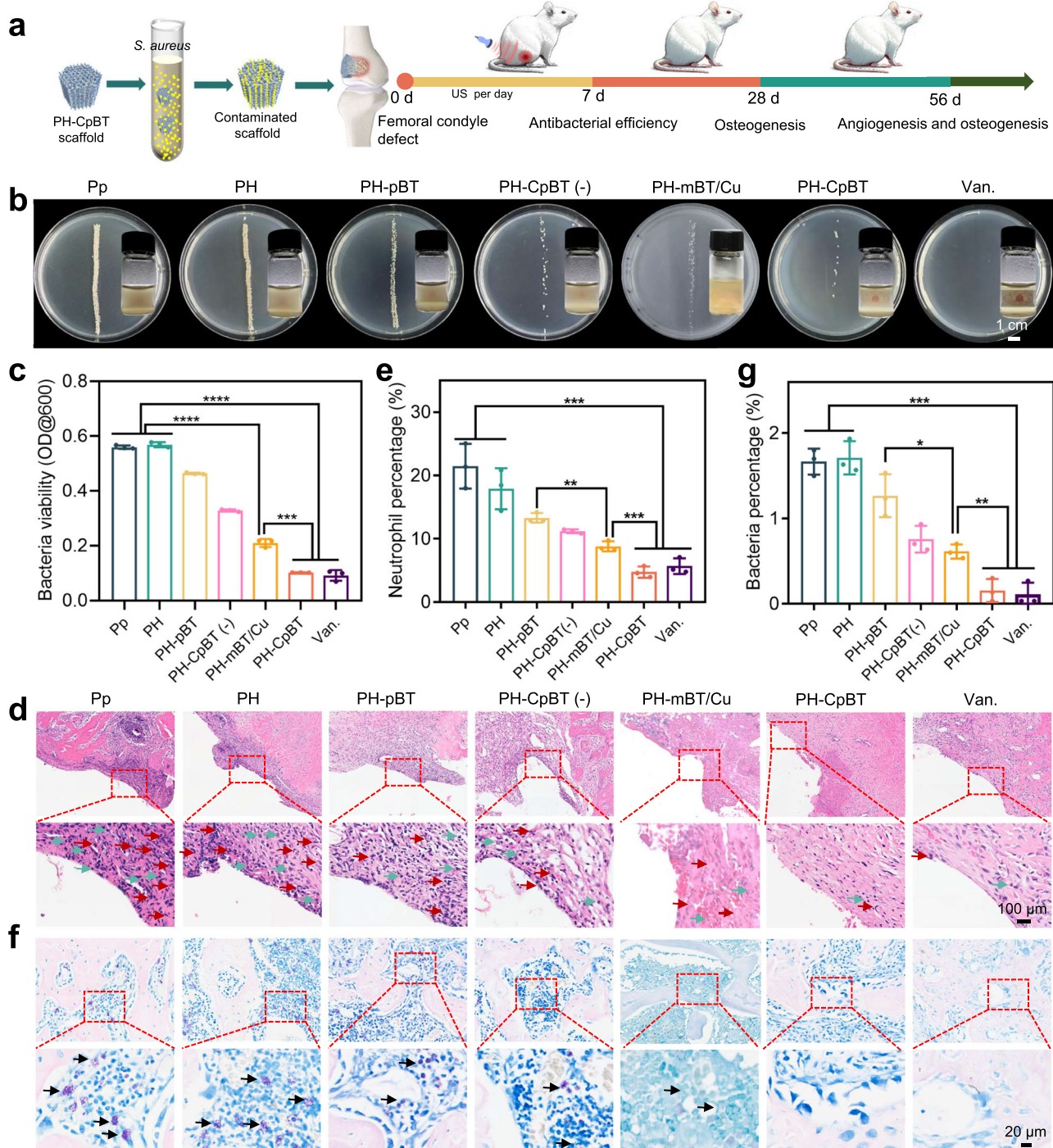

**Fig. 5 | The treatment of implant infection with PH-CpBT in vivo. a** Schematic illustration of infected modified PEKK scaffolds and animal experimental treatment in therapeutic. **b** Photographs of bacterial colonies and turbid liquid. **c** Quantitative analysis of bacterial turbid liquid by OD 600. A representative image of three biological replicates from each group was shown. **d** H&E images and **e** semi-quantification of neutrophil in the infected bone tissues surrounding the implants. The red arrows represented neutrophils, and the green arrows represented lymphocytes. A representative image of three biological replicates from each group was shown. **f** Giemsa staining images and **g** semi-quantification of bacteria in the infected bone tissues surrounding the implants. The black arrows represented bacteria. A representative image of three biological replicates from each group was shown. **c, e, g** $n = 3$ biologically independent samples; ANOVA followed by Tukey's multiple comparisons; data were presented as mean values ± SD; error bars = SD. Significant differences between groups were indicated as ****$p < 0.0001$, ***$p < 0.001$, **$p < 0.01$, and *$p < 0.05$. Source data are provided as a Source Data file.

the real-life scenario in vivo (Fig. 5a). On the 7th-day post-surgery, the animals were sacrificed to gather the femurs for bacteriological and histological analysis. Visual examination revealed the presence of secretions and pus at the implant site of Pp and PH, and partial mitigation were observed in PH-pBT, PH-CpBT (US-), and PH-mBT/Cu

(Supplementary Fig. 28). By comparison, the implant site of PH-CpBT +US and vancomycin (Van.) exhibited smooth tissue healing without secretion and pus formation, suggesting the clearance of bacterial infection. The order of bacterial colonies on agar plates and the turbidity of the Luria-Bertani (LB) medium after cultivation were as

follows: Pp ≈ PH > PH-pBT ≈ PH-CpBT (US-) > PH-mBT/Cu > PH-CpBT ≈ Van, suggesting that PH-CpBT exhibited favorable in vivo antibacterial characteristics rivaling Van (Figs. 5b and 5c). For PH-mBT/Cu, the lack of pDA as "electron aspirator" to initiate the reduction of $Cu^+$, the antibacterial performance by SDT and released $Cu^{2+}$ only was inefficient. For PH-pBT, no induction of copper-induced cuproptosis-like bacterial death occurred due to the absence of Cu ions, also exhibited moderate antibacterial effect. Hematoxylin and Eosin (H&E) revealed typical signs of bone tissue infection in Pp and PH, characterized by a large number of lymphocytes, monocytes, and neutrophil infiltrations into the tissues (Fig. 5d). Semi-quantitative analysis corroborated that PH-CpBT and Van. exhibited the least neutrophils, affirming efficient disinfection and minimal inflammatory response (Fig. 5e). Furthermore, Giemsa staining suggested the presence of numerous bacteria (Figs. 5f and 5g). Only a few bacteria were observed in PH-CpBT and Van., further supporting the effective bactericidal effect of PH-CpBT in in vivo environment and its potential for clinical sterilization.

## Bone regeneration in vitro and in vivo

The central objective of antibacterial treatment was to enhance bone ingrowth. To verify cell adhesion and proliferation, various scaffolds were cultured with MC3T3-E1 cells in vitro. Cell viability by Cell Counting Kit-8 (CCK-8) assay showed almost no cytotoxicity in all scaffolds (Supplementary Fig. 29). Besides, all scaffolds except for P and Pp showed similar proliferation rates (Supplementary Fig. 30). With 3 days of culture, cells efficiently infiltrated the entire volume of PH-CpBT, and their actin cytoskeleton exhibited an elongated morphology throughout the scaffold, indicating good biocompatibility and cell adhesion (Supplementary Fig. 31). SEM images (Supplementary Fig. 32) showed superior cell spreading and tight adhesion to PH-CpBT, and cells exhibited more extended filopodia than other scaffolds due to the excellent hydrophilicity of PH-CpBT. Following 7 and 14 days of osteogenic induction, activity was analyzed (Supplementary Fig. 33), suggesting the significant osteogenic differentiation of PH-CpBT. Similar findings were observed in Alizarin Red S (ARS) staining (Supplementary Fig. 34), where PH-CpBT exhibited a significant increase in calcium nodules after 14 and 21 days of incubation. The number of tube formations and junctions for tubulogenesis of human umbilical vein endothelial cells (HUVECs) was higher in PH-CpBT (Supplementary Fig. 35), underlining its advantageous angiogenic properties.

At 4th and 8th weeks post-implantation in vivo, the rat femur was scanned by micro-computed tomography (Micro-CT). 3D reconstructions of the femoral condyle and bone defect areas, facilitated by Imaris software, uncovered substantial bone loss around bone defects in Pp and PH, attributed to osteolysis induced by bacterial proliferation and spread (Supplementary Fig. 36). Conversely, PH-CpBT exhibited robust in vivo osteogenicity, evident from the extent of new bone formation (red section) (Fig. 6a, Supplementary Movie 1). Over time, new bone in PH-CpBT effectively filled the scaffold pores, including the middle of the scaffold. The presence of gaps in the top and front views of the new bone indicated the scaffold's suitability for bone growth. Although Van. exhibited similar antibacterial ability, PH-CpBT outperformed it by highlighting the good osteogenic ability. Based on micro-CT quantitative data, parameters such as bone volume/total volume (BV/TV), bone mineral density (BMD), trabecular number (Tb.N), trabecular separation (Tb.Sp), and trabecular thickness (Tb.Th) were most distinguished in PH-CpBT (Fig. 6b-f). This suggested that excellent antibacterial and osteogenic abilities were essential for PH-CpBT in vivo. To evaluate the rate of new bone deposition, calcitonin and alizarin red through intraperitoneal injection were used to mark the bone formation line. PH-CpBT exhibited the largest distance and area between the two calcium deposition lines (Fig. 6g). Quantitative analysis,

including mineral apposition rate (MAR), revealed that PH-CpBT stimulated faster bone deposition compared to the other groups (Fig. 6h).

## New bone ingrowth: histopathology, histochemistry, and immunofluorescence

The undecalcified implant-bone tissue underwent H&E staining and Goldner trichrome staining after 4th and 8th weeks of implantation (Fig. 7a). The H&E staining clearly showcased heightened new bone formation encircling the implant, alongside more pronounced bone ingrowth within the pores of PH-CpBT. Quantitative evaluation of H&E staining showed that PH-CpBT had a higher percentage of new bone area (60.1% at 8 weeks) compared to other groups and Van. (Fig. 7b). Goldner trichrome staining demonstrated PH-CpBT exhibited more new mineralized bone surrounding implant (Fig. 7a), and quantitative analysis confirmed highest amount of new tissues in PH-CpBT (Fig. 7c). These two staining demonstrated that bone ingrowth was primarily observed at the edge of the scaffolds, gradually penetrating deeper inside the porous scaffolds over time, and the fine structure of new bone resembled the surrounding normal bone at the scaffold edge, tightly attaching to PH-CpBT. Histological analysis of harvested organs (heart, liver, spleen, lung, and kidney) from the rats in PH-CpBT showed no organic changes, indicating satisfactory biosafety (Supplementary Fig. 37). Blood routine testing also revealed favorable indicators in the experimental group (Supplementary Fig. 38), confirming the satisfactory biocompatibility and minimal side effects of PH-CpBT. Moreover, immunohistochemistry demonstrated that the expression of tumor necrosis factor-α (TNF-α) decreased in PH-CpBT (Supplementary Fig. 39), confirming the remarkable antibacterial effect of PH-CpBT in eradicating infection and reducing inflammation in vivo, which contributed to accelerated bone regeneration and vascularization. The ingrowth of bone and blood vessels at the defect site was evaluated using four-color immunofluorescence (DAPI, CD31, BMP-2, and RUNX2) based on tyramide signal amplification (TSA). PH-CpBT exhibited the strongest fluorescence intensity (Fig. 7d). Immunofluorescence analysis pinpointed a higher density and fluorescence intensity of CD31 in PH-CpBT, signifying an augmented level of vascularization facilitated by copper (Fig. 7e). The activity of osteogenesis-associated proteins RUNX2 and BMP-2 exhibited distinct upregulation in PH-CpBT (Figs. 7f and 7g), indicating the implant's ability to promote bone formation, meeting the prerequisites for optimal bone growth within the implants. The upregulated expression of these osteogenesis-related factors can be attributed to the antibacterial ability of PH-CpBT under US and Cu loading, which effectively facilitated osteogenesis, demonstrating the significant osteogenic promotion of PH-CpBT. In general, this study provided a promising strategy for designing multifunctional bone implants with simultaneous highly effective antibacterial and osteogenic capacities based on US-activated tandem catalysis, offering a reference for future clinical applications.

## Methods

### Chemicals

Barium carbonate (99%), titanium dioxide (99%), calcium carbonate (99.9%), zirconium dioxide (99.99%), cupric sulfate (99%), and nano HA were from Sinopharm Chemical Reagent Co., Ltd. (China). Dopamine hydrochloride (DA-HCl), tri(hydroxymethyl) amino methane hydrochloride (Tris-HCl, 1 M), methylene blue (MB), vancomycin, Calcein, and ARS were from Sigma Chemical Co. (USA). SOSG was from Invitrogen (USA). The Bacterial Live/Dead Bac Light viability kit was from Thermo Fisher (USA). Calcein/PI Cell Viability/Cytotoxicity Assay Kit, CCK-8 kit, and ARS dye were purchased from Beyotime (China).

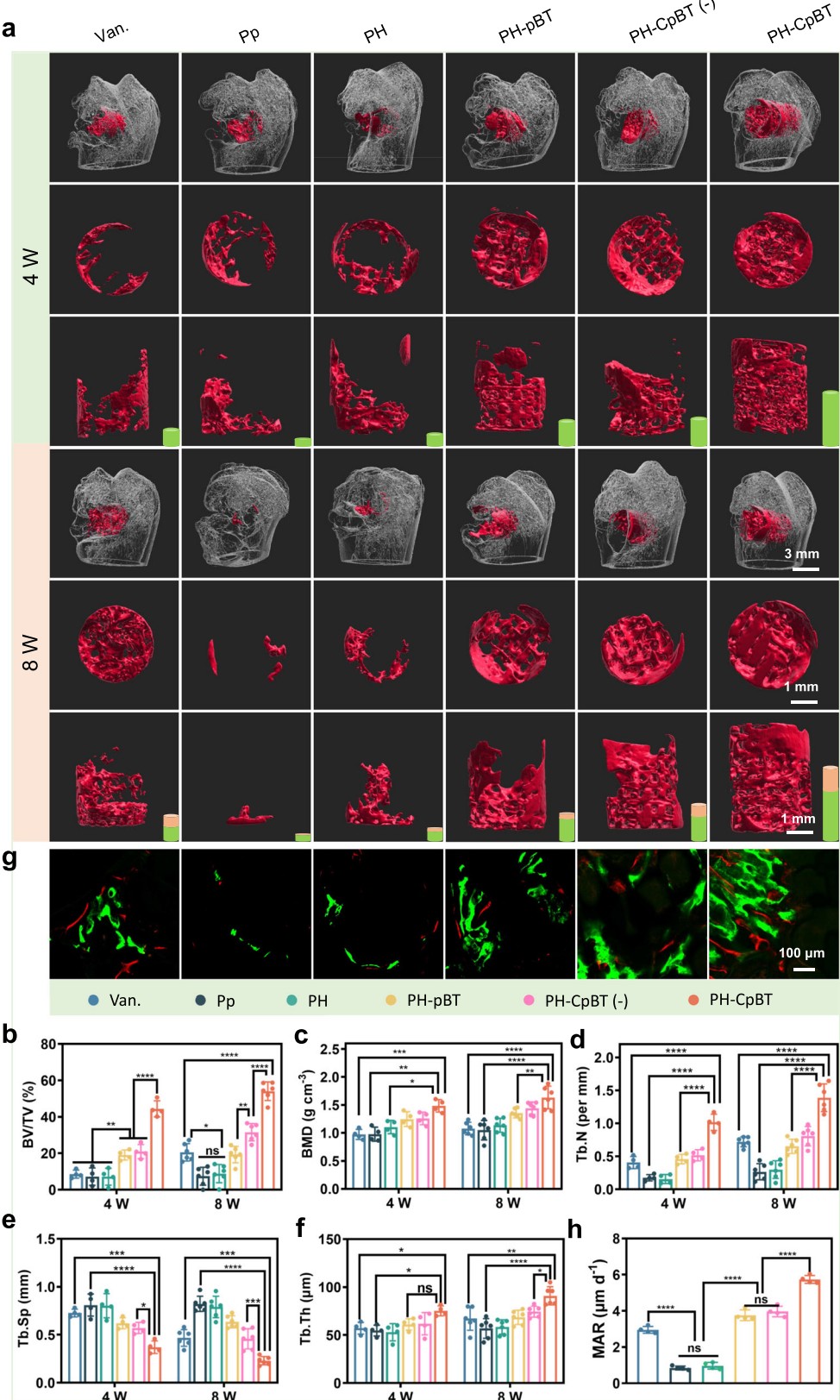

## Synthesis of CpBT nanoreactors

The mBT was prepared in the following steps: Weigh the elemental components according to the chemical formula $Ba_{0.90}Ca_{0.10}Ti_{0.91}Zr_{0.09}$. Ball-mill the weighed components in nylon jars for 24 hours. Calcined the obtained powders at 1260 °C for 3 hours. Sand-milled the calcined powders at 2000 revolutions per minute for 4 hours to obtain mBT. Then, 1 mg mL$^{-1}$ mBT was dispersed in the Tris-HCl solution (10 mM, pH = 8.5) and sonicated for 40 min. DA-HCl was added and sonicated for another 10 min and stirred for 2 hours. After centrifuging and washing with DI water for three times, pBT nanoparticles were obtained. Then,

**Fig. 6 | The number and speed of new bone growing into the scaffold. a** Mico-CT of femoral condyle, from up to down, reconstruction of the defect and new bone ingrowth in the scaffolds, top view of new bone, side view of new bone. The green and pink cylinder showed the new bone growth for the first four weeks and the next four weeks, respectively. A representative image of four (4 W) and six (8 W) biological replicates from each group was shown. Quantitative statistics of bone regeneration related index in 3D reconstruction by micro-CT including **b** BV/TV, **c** BMD, **d** Tb. N, **e** Tb. Sp, and **f** Tb. Th. **g** Calcitonin (green) and alizarin red (red) marked new bone. A representative image of three biological replicates from each group was shown. **h** Quantitative statistics of MAR from 4th week to 6th week. **b**–**f** $n = 4$ biologically independent samples (4 W), $n = 6$ biologically independent samples (8 W); **h** $n = 4$ biologically independent samples. **b**–**h** ANOVA followed by Tukey's multiple comparisons; data were presented as mean values ± SD; error bars = SD. Significant differences between groups were indicated as ****$p < 0.0001$, ***$p < 0.001$, **$p < 0.01$, and *$p < 0.05$. Source data are provided as a Source Data file.

10 uM cupric sulfate solution was added to pBT and stirred for 2 hours, and then were collected by centrifugation and washed with DI water for three times to obtain CpBT nanoreactor.

### Preparation of PH-CpBT scaffold

3D printing technology utilizing fused deposition modeling was employed to fabricate PEKK scaffolds. The design of the structures was accomplished using Materialise 3-Matic software, resulting in bone scaffolds measuring 3 mm in diameter and 4 mm in height for in vivo experiments, as well as 10 mm in diameter and 1 mm in height for in vitro experiments involving cells and bacteria. Medical-grade PEKK filaments were extruded into the deposition bin of the 3D printer, enabling layer-by-layer preparation of the scaffolds according to predetermined shapes at a temperature of 200 °C. The PEKK scaffold was stirred in Tris-HCl solution (10 mM, pH = 8.5) containing 3 mg ml$^{-1}$ DA-HCl for 24 hours to obtain PEKK@pDA (Pp). Then Pp scaffold was immersed in HA solution (2 mg ml$^{-1}$) for 12 hours to get Pp@HA (PH). The scaffolds of PH-pBT and PH-CpBT were constructed by soaking PH scaffolds in different solutions (2 mg ml$^{-1}$) for 12 hours.

### Characteristics

The crystal structures were unveiled through XRD utilizing Cu Kα radiation (λ = 1.5406 Å) (Empyrean, Malvern Panalytical, UK). The SEM (SUPRA 55, Carl Zeiss AG, Germany) was used to examine the surface morphologies. High-resolution portraits and the related lattice fringes were captured by HRTEM (Talos F200i, FEI, USA). Raman spectra were examined by Raman spectroscopy with an excitation source of 532 nm (Invia Reflex, Renishaw, UK). The chemical state and valence band were obtained by XPS (K-Alpha +, Thermo Fisher, USA). The switching spectroscopy piezo-response force microscopy loops were collected by a commercial atomic force microscope (MFP−3D, Asylum Research, UK). The temperature-dependent dielectric constant ($\varepsilon_r$-$T$) was obtained via an LCR meter (TH2816A, Tonghui, China). The 5982 Universal testing machine (Instron, USA) was used to perform the compression test. The contact angle of the surfaces of scaffolds was measured by A JY-82C contact angle apparatus (Dingsheng Testing Equipment Co. Ltd., China). Then, the concentration of Cu was obtained by an inductively coupled plasma-optical emission spectrometer (ICP-OES, model 5100, Agilent, USA). Cu$^{2+}$ ions released from PH-CpBT scaffolds were analyzed by an ICP spectrometer (ICP-MS, 7850, Agilent, USA). Specifically, PH-CpBT scaffolds were immersed in a 0.9 % NaCl solution at 37 ± 1 °C with the surface-area to -volume ratio was 3 cm$^2$ mL$^{-1}$ according to the international standard ISO 10993-12. Triplicate samples were used to obtain an average value with standard deviation.

### Detection of ROS in vitro

The ROS generated from the samples when exposed to US stimulation were tested using MB, SOSG, and ESR. To assess •OH production, different samples were mixed with a solution containing MB and subjected to US stimulation (1 MHz, 1.0 W cm$^{-2}$, 50% duty cycle) with or without H$_2$O$_2$ (50 μM). The changes in absorption of MB at 664 nm before and after US stimulation were recorded using the ultraviolet and visible spectrophotometer (UV-5200, METASH, China). Similarly, the presence of singlet oxygen was confirmed by the fluorescence intensity of SOSG at 525 nm using a fluorescence spectrophotometer

(F-7000, Hitachi, Japan). The identification of ROS species was performed using an ESR spectrometer (JES-FA200, JEOL, Japan). For trapping the singlet oxygen, we utilized 2,2,6,6-tetramethylpiperidine (TEMP) at a concentration of 50 mM, while for detecting •OH, we employed 5,5-dimethyl-1-pyrroline-N-oxide (DMPO) at a concentration of 0.1 mM.

### Detection of Cu$^+$ of CpBT

Neocuproine (Aladdin, China) was selected as an indicator for detecting in-situ Cu$^+$ production. Briefly, 1.04 mg of neocuproine was dissolved in 5 mL of ethyl alcohol and then diluted five times with ultrapure water. Subsequently, the buffer solution at pH 6.5 was prepared by dissolving KH$_2$PO$_4$ and NaOH in ultrapure water. CpBT dispersion at a concentration of 400 μg mL$^{-1}$ was prepared by dissolving itself in ultrapure water, and then the final working solution, consisting of 0.75 mL of buffer solution, 1.0 mL of CpBT solution, and 0.8 mL of neocuproine solution, was irradiated under sonication for 0-12 min (1 MHz, 1.0 W cm$^{-2}$, 50% duty cycle). Finally, the absorption of the reaction solution was measured by a UV-vis absorbance spectrometer at 452 nm.

### Simulation details

The dimer, trimers, and tetramers structures of pDA employed in the current study were extracted from the most stable geometry among all the structures generated using a brute-force algorithmic generator[37]. The layered aggregates consistent of the stacked dopamine (DA) and dihydroxyindole (DHI) via π-π interactions[31]. The pDA-Cu$^{2+}$/Cu$^+$ composites were constructed by chelating Cu$^{2+}$/Cu$^+$ ions near the dihydroxyl site[38,39]. Unless otherwise specified, the Becke three-parameter Lee-Yang-Parr (B3LYP) functional was employed to optimize the molecular structure and generate the wavefunction, together with the def2-TZVP basis set. The BLYP functional combined with the 6−31+g (d, p) basis set was employed to optimize the structures of the layered aggregates and analyze the wavefunction. To account for weak interactions in the layered aggregates systems, the dispersion correction DFT-D3 with Becke-Johnson damping (D3BJ) was implemented[40]. The Pipek-Mezey method[41] was chosen to localize occupied molecular orbitals (MOs) and differentiate σ and π characters. For investigating the delocalization channel of π electrons, the LOL-π variant of the localized orbital locator (LOL) function was utilized[27,28]. All quantum chemistry calculations were performed using the Gaussian16 program[42], while wavefunction analysis was conducted using the Multiwfn 3.8 (dev) code[43]. Isosurface graphs were generated using the VMD program for improved visual representation. To investigate the quantum transport properties of multilayer stacked pDA, we designed a vertical transport architecture device with Au electrodes being the source and drain contacts, and pDA was acted as the transport channel region. The simulations were performed using the first-principles methods, with the combination of the NEGF-DFT[29], conducted in the first-principles quantum transport software package *Nanodcal*[44]. The local density approximation (LDA) was embraced to portray the exchange and correlation function, and atomic cores were defined by the standard norm-conserving nonlocal pseudo potentials[45]. A double-ς polarized (DZP) atomic orbital basis[46] was conducted for Au metal electrode and pDA to expand all physical quantities with a kinetic energy cutoff of 4500 eV. Furthermore, a $k$-point mesh of $3 \times 3 \times 1$ and

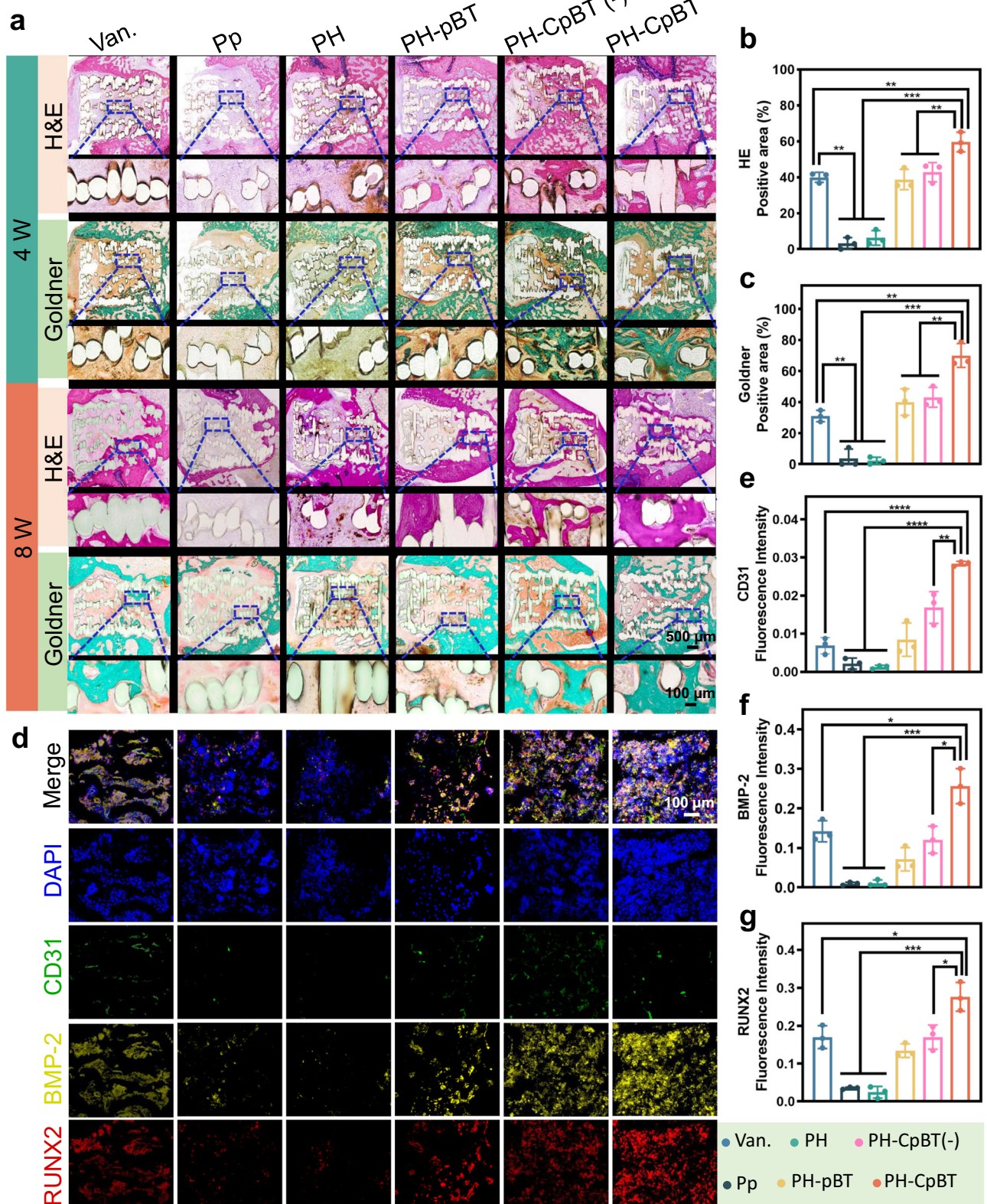

**Fig. 7 | The microstructure and mechanism of new bone ingrowth. a** H&E and Goldner staining of undecalcified tissue around the implant at 4th and 8th weeks post-surgery. A representative image of three biological replicates from each group was shown. **b** Quantification of H&E in the implants at 8th weeks. **c** Quantification of Goldner positive area in the implants at 8th weeks. **d** Immunofluorescence staining of CD31, BMP-2, and RUNX2 surrounding tissues of implants at 8th weeks post-surgery. A representative image of three biological replicates from each group was shown. Quantification of **e** CD31, **f** BMP-2, and **g** RUNX2 immunoreactivity within the implants. **b, c, e, f, g** $n$ = 3 biologically independent samples; ANOVA followed by Tukey's multiple comparisons; data were presented as mean values ± SD; error bars = SD. Significant differences between groups were indicated as ****$p$ < 0.0001, ***$p$ < 0.001, **$p$ < 0.01, and *$p$ < 0.05. Source data are provided as a Source Data file.

$17 \times 17 \times 1$ was applied to sample the first Brillouin zone for integrations in the reciprocal space of the scattering region (pDA) and Au electrode, respectively. In addition, self-consistent calculations were converged until each component of the density matrix have declined to $10^{-5}$ Hartree. Employing the Landau-Ginsburg-Devonshire phenomenological model, the Landau free energy was expressed as $\triangle G = \alpha_1(P_1^2 + P_2^2 + P_3^2) + \alpha_{11}(P_1^4 + P_2^4 + P_3^4) + \alpha_{12}(P_1^2 P_2^2 + P_2^2 P_3^2 + P_1^2 P_3^2) + \alpha_{111}(P_1^6 + P_2^6 + P_3^6) + \alpha_{112}[P_1^4(P_2^2 + P_3^2) + P_2^4(P_1^2 + P_3^2) + P_3^4(P_1^2 + P_2^2)] + \alpha_{123}P_1^2 P_2^2 P_3^2$, where $\alpha_1$, $\alpha_{11}$, $\alpha_{12}$, $\alpha_{111}$, $\alpha_{112}$ and $\alpha_{123}$ represented Landau energy coefficients. Finite element method calculations were carried out with COMSOL Multiphysics 5.4 with a model of a piezoelectric device based on a steady-state study.

### Culturing of bacteria
Gram-positive *S. aureus* (ATCC 25923) and Gram-negative *E. coli* (ATCC 25922) were cultured in a sterile Luria-Bertani (LB) medium (10 g L$^{-1}$ of back to-tryptone,10 g L$^{-1}$ of NaCl, and 5 g L$^{-1}$ of bacto-yeast extract). The bacterial counts were obtained from the spread plate of different samples.

### Antibacterial assessment in vitro
The effectiveness of Pp, PH, PH-pBT, and PH-CpBT against *S. aureus* and *E. coli* with and without US stimulation was assessed using the spread plate method. The different scaffolds were exposed to bacterial suspensions ($2 \times 10^7$ CFU mL$^{-1}$) in 48-well plates for specified durations and subjected to US stimulation (1 W cm$^{-2}$) for 9 mins or left untreated. Bacterial growth was cultured on agar plates at 37 °C for 18 h using the spread plate method to quantify colony-forming units (CFUs). To simulate the high H$_2$O$_2$ environment in vitro, PH-CpBT+H$_2$O$_2$ was supplemented with 50 μM H$_2$O$_2$. Furthermore, under identical conditions, antibacterial experiments were conducted to compare copper chelator (TTM, 20 μM, Aladdin, China), copper release from the scaffold, the coating of mBT and copper ions (PH-mBT/Cu), and similar materials (commercial TiO$_2$, ZnO, MoS$_2$, and BT, Aladdin, China). The adherent bacteria on PEKK were fixed, dried, dehydrated, and coated with gold. SEM (ZEISS Gemini 300, Germany) was employed to examine bacterial morphology and integrity. Bacteria treated with US and PH-CpBT+H$_2$O$_2$ + US were collected, cryo-centrifuged at high speed for 2 mins, embedded into blocks, and sliced into 50 nm thick sections using an ultrathin microtome (EM UC7, Leica, Germany). Subsequently, TEM images were captured after staining the samples with uranyl acetate and lead citrate, and placing on copper grids. After 5 days of bacterial cultivation on various scaffolds, a live/dead staining assay was conducted employing the Live/Dead BacLight viability kit. The data were collected by the confocal laser scanning microscopy (CLSM, N-SIM S, Nikon, Japan). Moreover, intracellular ROS levels were measured using the ROS assay kit (Beyotime, China) four hours after coculturing with different scaffolds, and the outcomes were also visualized using CLSM. The assessment of bacterial DNA damage induced by various samples utilized the Bacterial DNA Kit (Beyotime, China). After treatment, bacteria underwent collection through centrifugation at 5000× *g* for 5 mins at 4 °C. Following the manufacturer's instructions, the Bacterial DNA kit was employed for the purification of intact DNA strands from the bacteria. Subsequently, complete DNA fragments were subjected to quantitative analysis using a UV-vis spectrophotometer. This comprehensive methodology aimed to discern and quantify the extent of DNA damage caused by the diverse samples. The supernatant obtained by centrifugation was detected by BCA Protein Quantitation Assay (Beyotime, China), and the bacterial leakage protein was quantified by a microplate reader.

### Transcriptome analysis
*S. aureus* was cultivated with PH-CpBT with and without US stimulation. The bacteria were collected by cryo-centrifugation during the logarithmic phase and immediately frozen in liquid nitrogen. Three

sets of biological replicates were carried out under identical conditions: the control group (US-1, US-2, US−3), the experimental group 1 (PH-CpBT+H$_2$O$_2$-1, PH-CpBT+H$_2$O$_2$-2, PH-CpBT+H$_2$O$_2$−3), and the experimental group 2 (PH-CpBT+H$_2$O$_2$ + US-1, CpBT+H$_2$O$_2$ + US-2, CpBT+H$_2$O$_2$ + US−3). Total RNA isolation and cDNA library construction were executed according to the manufacturer's instructions, followed by sequencing on an Illumina HiSeq platform at Majorbio Biopharm Technology Co., Ltd. The expression quantification results were subjected to analysis using DESeq2 (Version 1.24.0) software to discern DEGs with a screening threshold of $|\log 2FC| \geq 1$ and a *p*-value < 0.05. We utilized Kyoto Encyclopedia of Genes and Genomes (KEGG, http://www.genome.jp/kegg/) databases to elucidate the biological implications of DEGs and explain gene functional differences between samples. Furthermore, we conducted KEGG pathway analysis by KOBAS. The significance of KEGG pathways was evaluated through a Fisher's exact test.

### Metabonomics analysis
The sample preparation for the metabolomic analysis was performed as described in transcriptome analysis, and metabolomics analysis was performed under the same conditions for each group of six biological replicates. The experimental procedure was as follows: preparation of the central carbon metabolite standard solution; pre-processing of the standard curve, treatment of the bacterial cell precipitate samples, and qualitative and quantitative detection of target compounds in the samples using LC-ESI-MS/MS (UHPLC-Qtrap) with the ExionLC AD system; Chromatographic analysis was performed using the Waters HSS T3 column (2.1 ×150 mm, 1.8 μm); Mass spectrometry analysis was conducted using the SCIEX QTRAP 6500+ in both positive and negative modes. Finally, the Sciex OS quantitative software was utilized for the automatic identification and integration of ion fragments to analyze the data.

### Biocompatibility assessment in vitro
Mouse osteoblastic MC3T3-E1 cells (GNM15) were purchased from the Cell Bank of the Chinese Academy of Sciences (Shanghai, China). The HUVECs (CP-H082) were purchased from Procell Life Science & Technology Co., LTD. (Wuhan, China). MC3T3-E1 cells cultured with α minimum essential medium (α-MEM, Gibcom, USA), including 10% fetal bovine serum (FBS, Gibcom, USA) and 1% penicillin-streptomycin solution. The cultures were maintained in a humidified atmosphere incubator with 5% CO$_2$ at 37 °C for in vitro cytocompatibility and osteogenic differentiation studies.

### Cell proliferation and spreading assay
Cell viability and proliferation were quantified through a CCK-8 assay. Various scaffolds were co-cultured with cells, and the optical density (OD) at 450 nm was gauged on days of 1, 3, and 5 using a microplate reader. Moreover, cells were seeded onto different scaffolds for 3 days, and their viability was gauged using a Calcein/PI Cell Viability/Cytotoxicity Assay Kit, followed by observation using laser scanning CLSM. MC3T3-E1 cells were seeded onto different scaffolds in a 24-well plate for 24 h, and the cells were stained with fluorescein isothiocyanate-phalloidin and 4,6-diamidino-2-phenylindole, then the arrangement of F-actin and the cell nuclei were then observed using CLSM.

### Osteogenic differentiation and angiogenesis in vitro
MC3T3-E1 cells were seeded on the surface of different scaffolds. After 24 h, the culture medium α-MEM was replaced with an osteoinductive medium containing 10 mM β-glycerophosphate, 50 μg mL$^{-1}$ ascorbic acid, and 10 nM dexamethasone (all from Sigma, USA) to prompt the osteogenic commitment of MC3T3-E1 osteoblasts. This osteoinductive medium was renewed every 3 days. On days 7 and 14, the activity of ALP was assessed using a 5-bromo-4-chloro−3-indolyl phosphate/nitroblue tetrazolium (BCIP/NBT) ALP color development kit (Beyotime, China). For the evaluation of calcified extracellular matrix, cells were

fixed and treated with ARS dye on days 14 and 21. Then subsequent to the removal of excess dye, images were captured using a scanner. HUVECs were cultured for in vitro angiogenesis investigation. The suitable cells and samples were introduced into the μ-Slide 15 well 3D (Ibidi, Germany) and allowed to incubate for 6 hours. Afterward, the cells were stained with Calcein and subsequently imaged using a fluorescence microscope.

## In vivo experiments

Male SD rats (200–220 g, about two-months old) were bought from the Beijing Huafukang Bioscience Cojnc. Animal experimentation in this study received ethical approval from the Laboratory Animal Ethics Committee of West China Hospital, Sichuan University (IACUC number 20221216022). All rats were raised in $25 \pm 3 \,^\circ C$ (temperature), 60-70% (humidity), and 12 h light/dark cycle conditions for two weeks before the experiments. SD rats ($n = 8$ per group) were divided randomly into seven groups: Pp, PH, PH-pBT, PH-mBT/Cu, PH-CpBT(-), PH-CpBT, and Van.. Except for PH-CpBT(-) and Van., other groups underwent ultrasound intervention. To create an implant-related contamination model, the engineered PEKK implants ($\oplus$ 3 mm × 4 mm) were immersed in S. aureus ($2 \times 10^7$ CFU $ml^1$) solution at 37 °C for 4 hours. Subsequently, a rat lateral femoral condyle model for cylindrical bone defect repair was created, and the implants with S. aureus were implanted in the bilateral femoral condyles. After 1st, 4th, and 8th weeks, the rats were sacrificed to assess the antibacterial effect and new bone formation.

## Antibacterial activity in vivo

Starting from the first to sixth days after surgery, rats were exposed to US stimulation (1 MHz, 1.0 W $cm^{-2}$, 50% duty cycle) for 8 mins under gas anesthesia induction. On the seventh day post-surgery, the rats were sacrificed, and the implants were removed and placed in sample collection tubes containing PBS to collect bacteria. Dilutions were cultured on agar plates to assess implant infection. Moreover, bone tissue around the scaffolds was gathered for histological analysis, including H&E and Giemsa staining. These sections were observed and captured using an inverted microscope (Olympus BX53, Japan).

## The volume of new bone and bone growth rate

To evaluate the volume of new bone and trabecular thickness within the implants, femoral condyles were scanned using the Quantum GX Micro CT (PerkinElmer, USA). The 3D reconstruction of the CT images was generated by Imaris 9.9 (BitPlane, Oxford Instruments). Following reconstruction, parameters such as BV, Tb. Th, % BS/BV, % BV/TV, Tb. N, and BMD were quantified using Skyscan NRecon software. For the assessment of new bone growth rate, ARS (30 mg $kg^{-1}$) and Calcein (20 mg $kg^{-1}$) were intraperitoneally injected into the rats at 4th and 6th week after implantation. Infected rats were euthanized at 8th week post-surgery, and their implants and surrounding bone were fixed, sliced, and examined using CLSM.

## The microstructure of new bone ingrowth: histomorphometry, immunohistochemistry, and immunofluorescence

Histological sections were generated parallel to the long axis of the implants around the undecalcified femoral condyle before decalcification. Sections were ground to 100 μm thickness, and slides were polished and stained with H&E and Goldner. Additionally, other bone samples without implants were fixed, decalcified, dehydrated, and embedded in paraffin. The tissues were cut into sections, being prepared for IHC staining (TNF-α, 1:200, Abcam ab1793) and 4-color immunofluorescence staining with the following primary antibodies (Abcam, UK): CD31 (1:500, ab182981), RUNX2 (1:200, ab236639), BMP-2 (1:200, ab214821), and DAPI (1:1000, ab285390) at 4 °C overnight. Finally, 4-color immunofluorescence staining was performed and visualized with Vectra Polaris (PerkinElmer, USA).

## Statistics analysis

Data were presented as mean values ± standard deviations (SD); error bars = SD. Statistical analysis was performed using ANOVA followed by Tukey's multiple comparison with statistical significance assigned at $*P < 0.05$, $**P < 0.01$, $***P < 0.001$ and $****P < 0.0001$.

## Reporting summary

Further information on research design is available in the Nature Portfolio Reporting Summary linked to this article.

## Data availability

The authors declare that all data supporting the findings of this study are available within the article and the Supplementary Information. The RNA-seq data generated in this study are available on the National Center for Biotechnology Information (NCBI) database under the BioProject PRJNA1064643. Any additional requests for information can be directed to, and will be fulfilled by, the corresponding authors. Source data are provided with this paper.

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

## Acknowledgements

Authors gratefully acknowledge the supports of the National Natural Science Foundations of China (D. W. No. 82102542, Y. L. H. No. 12204104, Z. K. Z No.82172394), the fellowship of China Postdoctoral Science Foundation (D. W. 2021M692283), the Natural Science Foun-dation of Fujian Province (Y. L. H. No. 2022J01631). Authors thank Li Li, Fei Chen, Chunjuan Bao, and Yang Deng (lnstitute of Clinical Pathology, West China Hospital) for processing histological staining; thank Lei Wu, Qiuxiao Shi, Yaping Wu and Li Zhou (Core Facility of West China Hospital, Sichuan University) for the help of micro-CT; thank Minghua Zhang (College of Polymer Science and Engineering, Sichuan University) for TEM micrographs; thank Xiaofeng Zhou (Strait Institute of Flexible Electronics, Fujian Normal University) for the help of the finite element method calculations.

## Author contributions

Y.H., X.W, and D.W. designed the research, performed the experiments, and wrote the manuscript. Q.S., C.Z., J.C., Y.Y., S.L., and X.C. performed the experiments and analyzed the data. J.Y. did the Landau free energy modeling. Y.D. discussed the results. T.W. did the computational simu-lation and discussed the results. X.Z. and Z.Z. supervised the research. All authors discussed the results and commented on the manuscript.

## Competing interests

The authors declare no competing interests.
