## [Peer Review File · Nature Communications]

REVIEWER COMMENTS

Reviewer #1 (Remarks to the Author):

The manuscript introduced a strategy utilizing US-activated tandem catalysis to amplify the therapeutic effect of SDT and CDT for the on-demand treatment of IAI via CpBT nanoreactors. The authors did a decent job in material design and biological experiments. However, the following issues must be clarified before accept this manuscript.

1. The temperature dependence of relative permittivity of the dielectric constant of CpBT is missing in Fig. 2e.
2. In Fig. 2f, the authors compare the temperature dependence of the dielectric constants of BT and mBT. The free-energy profiles of BT and mBT at different temperatures should be added and discussed to highlight the advantages of mBT.
3. In Fig. 2h, the peaks of Cu⁺ and Cu²⁺ in Cu 2p orbitals should be analyzed and the specific ratio of different valence states of copper before and after the reaction should be added as mentioned in the article. In addition, the colors of fresh and 934.9 eV are too similar and should be replaced.
4. A point should be added below the Lumo track in Fig. 3c and 3d to analyze the spatial scattering state and to prove that there is no transmission from the track below the Lumo track.
5. The article mentions that intracellular copper overload restricts the tricarboxylic acid cycle and promotes bacterial death by copper toxicity, so is it also cytotoxic in that state. Meanwhile, it is mentioned that the CpBT nanoreactor can cause DNA and protein leakage, but there is a lack of relevant proof.
6. In Fig. 4 d, the graph only gives US and US+H₂O₂+CPBT, and does not prove the effect of H₂O₂ alone on bacteria, the conclusion given in this experiment is not accurate enough.
7. The genes in Fig. 4g and 4h that are significantly different and associated with biofilm elimination should be labeled.
8. The characterization of material properties proves the conversion of copper ions, but no proof is given whether copper ions can play a role in in vivo experiments.
9. The antimicrobial performance or biofilm elimination ability of this nanoreactor CpBT should be compared with similar materials to demonstrate the advantages of this material.
10. In the in vivo experiments, the authors did not record the weight of each group of mice statistically, so there is no guarantee that each group of mice is basically the same.
11. It is suggested that the relevant content of the H&E images should be labelled, such as lymphocytes, and the quantitative analysis should be added for better observation and understanding by readers.
12. Statistical analysis should be added to the CT range area in Fig. 6a to visualize the speed of new bone growing.

13. DAPI is a DNA dye that clearly shows the nucleus and thus the distribution of cells, and should be the largest of all stains, while the range of DAPI in the stained picture in Fig. 7d is too small.

14. There is no description of the statistical analysis in the figure notes to Figs. 4 and 7, and there is no specific p-value range for **** in Figs. 5 and 6.

15. Fig. S1, Fig. S12 and Fig. S26 have problematic figure notes.

16. No scale bars have been added to Fig. S30.

Reviewer #2 (Remarks to the Author):

Huang et al. have developed a novel barium titanate-composite antibacterial coating for PEKK scaffolds. This coating employs a unique Tandem Catalysis mechanism, combining SDT and CDT to effectively combat bacterial infections. While their work is substantial, several significant shortcomings have led me to conclude that it may not be suitable for publication in Nature Communications. The specific issues are outlined below:

1. The author mentioned that the Cu^{2+} and single SDT is not active. However, numerous studies have demonstrated the potential of SDT in both tumor and bacterial eradication. Additionally, Cu^{2+} ions are widely recognized for their antibacterial properties in tissue engineering. Therefore, the authors should provide a more compelling explanation in their manuscript, addressing my primary concern: whether the notable antibacterial effect arises from an enhanced release of Cu^{2+} ions induced by ultrasound.

2. Following above, they must test the ion valence state of Cu in this reaction or the solution. The real anti-bacterial mechanism should be better revealed.

3. In the process of preparing the coated scaffold, the authors solely employ the method of solution soaking. This prompts the question of whether the antibacterial effect primarily results from ion release in a water-based environment. The stability of the coating under such conditions warrants investigation.

4. Furthermore, the authors have only presented the antibacterial properties of the powder form of their material. It is essential to assess the catalytic ability of the scaffold itself, as this will provide a more comprehensive understanding of its antibacterial potential.

5. Additionally, some studies also reported the potential for ultrasound to facilitate the conversion of Cu^{2+} ions into Cu^{+} ions. Consequently, it is imperative to reconsider and conduct further experiments to explore the relevance of SDT and CDT in this work. How important role of SDT play in this process?

Reviewer #3 (Remarks to the Author):

Huang et al. present a report on the design and anti-bacterial function of US-activated piezo-hot carriers. The study revealed that the carrier could induce the valence state interconversion between Cu^{2+} and Cu^{+} , and amplify the ROS generation via Cu^{+} -catalyzed chemodynamic reactions or copper overload-induced interruption of the tricarboxylic acid cycle. The manuscript is well-written and contains substantial data. The authors mentioned that metabolic programming played a role in the anti-bacterial function of the carrier, however, the analysis of data from metabolomics was missing. I think it is

necessary to include the data from metabolomics in the paper or supplementary information, as suggested below.

1.The author should clearly describe the differential metabolites in Control, PH-CpBT+H₂O₂, and PH-CpBT+US, as they did in Fig4.K.

2.It would be preferable if the author could perform a correlation analysis between differential metabolites and differentially expressed genes, the analysis will give the reader a comprehensive view of signal transduction and regulation, gene expression, and dynamic metabolite changes.

Detailed Response to the Reviewers' Comments

(No. NCOMMS-23-37813A)

Dear Reviewers,

Thank you for the constructive and insightful comments on our manuscript (No. NCOMMS-23-37813A) entitled “Ultrasound-Activated Piezo-Hot Carriers Trigger Tandem Catalysis Coordinating Cuproptosis-Like Bacterial Death Against Implant Infections”. We do appreciate the valuable comments and suggestions from the reviewers, which improve the quality of our manuscript. We have carefully revised the manuscript according to the reviewers’ suggestions. The annotations stated in this response letter and revisions made in the Revised Manuscript are listed as follows:

Reviewer #1

Comments:

The manuscript introduced a strategy utilizing US-activated tandem catalysis to amplify the therapeutic effect of SDT and CDT for the on-demand treatment of IAI via CpBT nanoreactors. The authors did a decent job in material design and biological experiments. However, the following issues must be clarified before accept this manuscript.

Answer: Thanks very much for your affirmation and comments. We have revised our manuscript and responded to your questions carefully. Specific explanations are shown below.

1. The temperature dependence of relative permittivity of the dielectric constant of CpBT is missing in Fig. 2e.

Answer: Thank you for your comment. The temperature dependence of the dielectric constant of CpBT was shown in **Figure R1**. The dielectric properties showed diffused phase transition at the temperature range of 20 to 45 °C due to the presence of polymer component pDA. In this work, the core mBT acted as an electron provider under US stimulation, and the shell pDA acted as an “electron aspirator” to transport electrons to Cu²⁺. Therefore, the dielectric and piezoelectricity properties of mBT were more important. In the manuscript, we paid more

attention to the dielectric properties of mBT, to clarify that in the body temperature mBT showed better piezoelectric performance than pure BT, which was in accordance with Landau free energy modeling.

Figure R1 The temperature dependence of dielectric constant of CpBT measured at 1 kHz.

2. In Fig. 2f, the authors compare the temperature dependence of the dielectric constants of BT and mBT. The free-energy profiles of BT and mBT at different temperatures should be added and discussed to highlight the advantages of mBT.

Answer: Thank you for your comment. We have conducted the Landau free energy modeling of pure BT at different temperatures (**Figure R2**). The pure BaTiO₃ in the T phase showed large polarization anisotropy energy ($> 1 \text{ J/cm}^3$) in the temperature range of 25-43 °C. However, in the same temperature range, mBT showed significantly decreased polarization anisotropy with a small energy gap among O and T phases (**Figure R3**) ($< 0.3 \text{ J/cm}^3$), favoring the higher piezoelectricity.

Figure R2 The free-energy profiles for pure BT at different temperatures.

Figure R3 The free-energy profiles for mBT at different temperatures.

We have added the Landau free energy profiles of pure BT at 37 °C to Figure 2f, and more discussion has been made in the Revise Manuscript on page 7, line 22:

“Additionally, the Landau free energy modeling revealed that at temperature of 37-43 °C, the polarization anisotropy energy and energy barrier ($< 0.3 \text{ J/cm}^3$) were relatively lower than that of pure BT ($> 1 \text{ J/cm}^3$) (Fig. 2f, Supplementary Fig. 3 and Fig. 4), which led to a small energy

barrier for polarization rotation among T <100> and O <110> states²⁶, hence inducing the enhanced piezoelectric performance over the body temperature.”

3. In Fig. 2h, the peaks of Cu⁺ and Cu²⁺ in Cu 2p orbitals should be analyzed and the specific ratio of different valence states of copper before and after the reaction should be added as mentioned in the article. In addition, the colors of fresh and 934.9 eV are too similar and should be replaced.

Answer: Thank you for your comment. The peaks of Cu⁺ and Cu²⁺ in Cu 2p orbitals before reaction had been shown in Supporting Information before. Now, we have added it to Figure 2i, and the color of the peak at 934.9 eV had been replaced. It showed that the higher peak at ~935 eV in Cu 2p_{3/2} spectra was assigned to Cu²⁺, accompanied by the characteristic Cu²⁺ shakeup satellite peaks (938-945 eV). The lower peak at ~932 eV suggested the presence of Cu⁺ or Cu⁰ species. Furthermore, the Cu LMM Auger spectra at ~570 eV confirmed the presence of Cu⁺ after US. Notably, the integral area ratio of Cu⁺ to Cu²⁺ after US was significantly enhanced at 935 eV (from 0.28:1 for fresh CpBT to 0.67:1 for used CpBT) and at 570 eV (from 0.33:1 for fresh CpBT to 0.5:1 for used CpBT) (Fig. 2j), indicating that part of surface Cu²⁺ species were reduced to Cu⁺ species during US stimulation.

Fig. 2 i XPS of Cu 2p_{3/2} for CpBT before and after US stimulation (the insets show Cu LMM spectra).
i Peak ratio of Cu⁺ to Cu²⁺ at 570 eV and 932 eV in XPS spectra of CpBT before and after US stimulation.

4. A point should be added below the Lumo track in Fig. 3c and 3d to analyze the spatial

scattering state and to prove that there is no transmission from the track below the Lumo track.

Answer: We are grateful to the reviewer for their meticulous examination of our manuscript and their insightful comments. Following the reviewer's suggestion, we have incorporated the real-space scattering states of the 16-layer stacked DHI and DA system below the LUMO energy level (< 0.1 eV) into Fig. 3c and 3d (as depicted below). The absence of a transmission coefficient in the transmission spectra below the LUMO energy level, along with the closed transport channel observed in the real-space scattering states, clearly indicated there was no occurrence of transmission within stacked DHI and DA system at these energy levels.

Fig 3. **a** Energy levels and **b** the corresponding density of states (DOS) of 16-layer stacked DHI and DA. **c** The transmission spectra of the 16-layer stacked DHI and DA-based transport architecture device. **d** The real space scattering states of the 16-layer stacked DHI and DA system at different energy levels. All the figures share the same color bar.

In manuscript, page 11, line 7: “Interestingly, the transmission spectra (Fig. 3c and 3d) revealed that the transmission coefficient near HOMO and LUMO was zero, and that transmission cannot occur at these energy levels, consistent with the LOL- π isosurfaces observations in the layered aggregation structure that there was no overlap of interlayer charge densities (Supplementary Fig. 7).” in the Results and discussions section was changed to “Notably, the transmission spectra (Fig. 3c) demonstrated a zero transmission coefficient near the HOMO and LUMO, with no observable opened transport channel in the real-space scattering states below

the LUMO energy level (< 0.1 eV, 2 in Fig. 3d). The absence of transmission around the HOMO and LUMO energy levels concurred with the observations of a non-overlapping interlayer charge density in LOL- π isosurfaces within the layered aggregation structure.”

5. The article mentions that intracellular copper overload restricts the tricarboxylic acid cycle and promotes bacterial death by copper toxicity, so is it also cytotoxic in that state. Meanwhile, it is mentioned that the CpBT nanoreactor can cause DNA and protein leakage, but there is a lack of relevant proof.

Answer: Thank you for your comment. In the human body, copper served a diverse range of functions. Under normal physiological conditions, intracellular copper concentrations were kept at extraordinarily low levels by active homeostatic mechanisms that worked across concentration gradients to prevent the accumulation of free intracellular copper that was detrimental to cells [*Nat. Chem. Bio.*, 2008, 4(3): 176-185; *Nat. Rev. Cancer*, 2022, 22(2): 102-113]. To figure out whether Cu ions had toxicity in this work, Cu ions released from PH-CpBT scaffolds at 1-, 4-, and 7-day intervals and with or without US stimulation were analyzed by an inductively coupled plasma (ICP) spectrometer according to the international standard ISO 10993-12, as shown in **Figure R4a**. It showed that PH-CpBT scaffold released Cu ions at a concentration of 3-7 $\mu\text{g/L}$ with or without US stimulation. The cytotoxicity of Cu ions was closely related to the concentration and cell lines, and **Table R1** summarized the toxicity of Cu ions concentration for different cells. It can be easily found that a Cu ions concentration < 6.3 mg/L was safe for lots of cell lines, which demonstrated that Cu ions released from PH-CpBT scaffolds (3-7 $\mu\text{g/L}$) would not cause cytotoxicity. Besides, based on our experiment, PH-CpBT scaffolds showed no cytotoxicity to MC3T3-E1 cells (**Figure R4b**). Moreover, it was also reported that Cu ions in a low concentration could accelerate the osteoblasts proliferation and the cell differentiation [*Bioact. Mater.* 2023, 23, 101–117; *Biomaterials* 2013, 34(2) 422–433], and significantly promote the proliferation of Vascular endothelial cells (VECs) [*Biomaterials* 2021, 268, 120553; *Biomaterials* 2022, 288, 121751]. As a result, the Cu element on PH-CpBT scaffolds showed no cytotoxicity.

Besides, in the bacteria-infected microenvironment, enriched H_2O_2 (ca. μM) was a crucial

factor for the generation of substantial ROS, which increased bacterial membrane permeability, causing the deactivation of Cu-related regulatory systems. Therefore, in the presence of H₂O₂, CpBT nanoreactor can generate ROS to facilitate copper entering bacterial cells, thus inducing copper toxicity, thereby achieving bactericidal effects. However, in a normal cellular environment with low H₂O₂ levels (ca. nM) [*P. Natl. Acad. Sci. USA*, 2017, 114(21): 5343-5348; *Angew. Chem. Int. Edit.* 2023, 62(7): e202210415], the cellular copper toxicity was not activated.

Figure R4 a Cu ions released from the PH-CpBT scaffold in 0.9 % NaCl solution with different immersion time and US stimulation at 37 °C. b Cytotoxicity of different scaffolds to MC3T3-E1 cells in this work.

Table R1 Cytotoxicity of Cu ions reported in literatures

Cu ion concentration (µg/L)	Cell line	Toxicity	Reference
63	Rabbit osteoclast	No	Prog. Nat. Sci. 2003, 13(4), 266–270.
504	Human osteoclasts	No	Int. J. Mol. Sci. 2021, 22, 2451
6-63.5	Mouse osteoblasts	No	Chinese J. Inorg. Chem. , 2010, 26(12): 2251-2258.
6.3×10^3	Human MG-63 osteoblasts	No	J. Biomed. Mater. Res. , 2002, 60(3), 420–433.
6.3×10^3	Mesenchymal stem cells	No	Mater. Sci. Forum , 2010, 638-642, 600–605.
3.175×10^3	Human endometrial epithelial cells	No	Contraception , 2012, 85, 509–518

The bacterial protein leaking from *S. aureus* after incubation on different scaffolds was

determined by the Micro BCA protein assay reagent kit, as shown in **Figure R5a**. PH-CpBT with US stimulation had much higher total proteins than other groups, confirming that cell membranes were damaged seriously. Besides, the total intact bacterial DNA was also obtained by Bacterial DNA Kit for different groups, and the quantitative data was shown in Figure R5b. Of note, only 20% of intact DNA was detected in PH-CpBT+H₂O₂+US group, suggesting that bacterial DNA was gravely devastated by the US-initiated ROS storm, which was consistent with transcriptomics (Fig. 4g-k).

Figure R5 Quantitative analysis of **a** protein leakage and **b** the intact bacterial DNA of *S. aureus* on different scaffolds after different treatments.

We have added more discussion to the Manuscript on page 15, line 4:

“The leaking of bacterial proteins was detected using the protein leakage assay, and PH-CpBT exhibited the highest protein leakage, indicating significant membrane permeability alteration and cytoplasmic leakage after US-activated tandem catalysis (Supplementary Fig. 22a). Besides, the total bacterial DNA was quantified by Bacterial DNA Kit for different groups (Supplementary Fig. 22b). Notably, only a few intact DNA was detected in PH-CpBT+H₂O₂+US, indicating severe damage to bacterial DNA due to the initiated ROS storm.”

About Cu-related cytotoxicity, more discussion has been added to page 22, line 3:

“To verify cell adhesion and proliferation, various scaffolds were cultured with MC3T3-E1 cells in vitro. Cell viability by CCK-8 assay showed almost no cytotoxicity in all scaffolds (Supplementary Fig. 29). Besides, all scaffolds except for P and Pp showed similar proliferation

rates (Supplementary Fig. 30). With 3 days of culture, cells efficiently infiltrated the entire volume of PH-CpBT, and their actin cytoskeleton exhibited an elongated morphology throughout the scaffold, indicating good biocompatibility and cell adhesion (Supplementary Fig. 31).”

6. In Fig. 4d, the graph only gives US and US+H₂O₂+CpBT, and does not prove the effect of H₂O₂ alone on bacteria, the conclusion given in this experiment is not accurate enough.

Answer: Thank you for your comment. As the reviewer suggested, to explore the influence of H₂O₂, we included H₂O₂+US and CpBT+US groups, as shown in Figure R6. It can be observed that when bacteria with the addition of H₂O₂ was subjected to US stimulation, both *S. aureus* and *E. coli* maintained intact morphology. In contrast, CpBT+US group exhibited only partially blurry bacterial membrane structures. In the presence of bacteria, the elevated levels of H₂O₂ in the vicinity were a result of the host's defense mechanism against the bacteria [*Clin. Micro. Rev.* 1997, 10(1): 1-18]. In order to simulate the elevated H₂O₂ environment in the infected microenvironment, we utilized the same concentration of H₂O₂ as in previous studies (*Angew. Chem. Int. Ed.* 2023, 62, e2022104). Consistent with their findings, our study also concluded that the involvement of H₂O₂ alone will not result in significant structural damage to bacterial cells.

Figure R6 TEM images of bacteria after different treatments.

We have added more discussion to the Manuscript on page 14, line 8:

“To observe microscopic changes in dead bacteria, Bio-TEM images of bacteria were examined

(Supplementary Fig. 17a). Bacteria incubated with PH-CpBT+H₂O₂+US showed incomplete walls and cytoplasmic leakage in both *E. coli* (Fig. 4d) and *S. aureus* (Fig. 4e), and bacteria with H₂O₂+US maintained intact morphology.”

7. The genes in Fig. 4g and 4h that are significantly different and associated with biofilm elimination should be labeled.

Answer: Thank you for your comment. To comprehensively understand the changes in biofilm-related genes across the three groups, we marked these genes on the heatmap (Fig. 4k). This visualization enabled a clearer examination of the differences in gene expression levels among the three groups. It was evident that PH-CpBT significantly interfered with biofilm formation.

Fig. 4 k Left, heat map showing the differential expression of genes of interest. Right, the death pathways by ROS-induced oxidative stress and Cu-induced cuproptosis-like death.

We have added more discussion to the Manuscript on page 16, line 24:

“In results, CpBT achieved the destruction of bacterial biofilm, induced DNA damage and proteins leakage of bacteria through the toxicity of ROS-related oxidative stress, and the inhibition of TCA cycle by cuproptosis-like effects, ultimately culminating incomplete sterilization.”

8. The characterization of material properties proves the conversion of copper ions, but no proof is given whether copper ions can play a role in in vivo experiments.

Answer: Thank you for your comment. As we proved that ultrasound-activated piezo-hot carriers triggering the reduction of Cu^{2+} to Cu^+ in vitro, which strengthened CDT and cuproptosis-like bacterial death, played an important part in antibacterial performance. During this process, pDA acting as an electron transport channel, was a key factor in the reduction of Cu^{2+} . Therefore, we fabricated Pp scaffolds coating with mBT and Cu ions (the content of Ba and Cu element was the same as PH-CpBT), defined as PH-mBT/Cu, as shown in Fig. 4. In this group, Cu^{2+} on PEEK failed to be reduced to Cu^+ in the absence of pDA on mBT, and the antibacterial effect originated from ROS generation by mBT and the released Cu ions. The animal experiment was conducted using PH-mBT/Cu group with US stimulation as we previously did, and the results were shown in Fig. 4. This approach proved less effective antibacterial performance (with the antibacterial rate of ~70%) than PH-CpBT, emphasizing the role of Cu ions conversion in enhancing CDT and cuproptosis-like bacterial death. Besides, PH-pBT, which lacked of Cu ions, also showed poor antibacterial results (<50%) with US stimulation, supporting the assertion that CDT and SDT in isolation have constrained antibacterial effects when compared to their combined effectiveness in PH-CpBT.

Besides, to elucidate it clearly and logically, a series of in vitro antibacterial experiments concerning Cu ions were conducted, as shown in **Figure R7**. Firstly, *S. aureus* or *E. coli*. were co-incubated with the leaching solution of PH-CpBT scaffold after US stimulation (30 min) for 1 day (Figure R7a), finding that the antibacterial rate was lower than 50% (Figure R7e-g), indicating that the antibacterial performance by released Cu ions only was not that good. Thirdly, to exclude the effect of released Cu ions on antibacterial effect, tetrathiomolybdate, a copper chelator [*Science* 2022, 375(6586): 1254-1261; *Dalton Trans.* 2022, 51(27): 10361-10376; *J. Neurol. Sci.* 1973, 20, 63-72], was co-incubated with bacteria and PH-CpBT (Figure R7b) under US stimulation. In this situation, Cu ions cannot get into the bacteria, so the antibacterial effect only came from ROS production by tandem catalysis between SDT and CDT, representing an antibacterial rate of ~70%, which indicated that Cu-related cuproptosis-like bacterial death was

also an important antibacterial mechanism in this work. At last, the same as in vivo experiment, PH-mBT/Cu also showed inefficient antibacterial performance, proving that the conversion of Cu^{2+} to Cu^+ was crucial in antibacterial performance.

Fig. 5 The treatment of implant infection with PH-CpBT in vivo. **a** Schematic illustration of infected modified PEKK scaffolds and animal experimental treatment in therapeutic. **b** Photographs of bacterial colonies and turbid liquid. **c** Quantitative analysis of bacterial turbid liquid by OD 600. **d** H&E images and **e** semi-quantification of neutrophil in the infected bone tissues surrounding the implants. The red arrows represented neutrophils, and the green arrows represented lymphocytes. **f** Giemsa staining images and **g** semi-quantification of bacteria in the infected bone tissues surrounding the implants. The

black arrows represented bacteria. The experiment was repeated three times independently, with a representative example shown. Significant differences between groups were indicated as **** $p < 0.0001$, *** $p < 0.001$, ** $p < 0.01$, and * $p < 0.05$.

Figure R7 a-d Schematic diagram for different antibacterial experiments. **e** Typical images of *S. aureus* and *E. coli* colonies treated by various groups. The corresponding antibacterial rate against **f** *S. aureus* and **g** *E. coli* after different treatments. The experiment was repeated three times independently, with a representative example shown. Significant differences between groups were indicated as **** $p < 0.0001$, *** $p < 0.001$, ** $p < 0.01$, and * $p < 0.05$.

We have added more discussion to the Manuscript on page 14, line 7:

“To observe microscopic changes in dead bacteria, Bio-TEM images of bacteria were examined (Supplementary Fig. 17a). Bacteria incubated with PH-CpBT+H₂O₂+US showed incomplete walls and cytoplasmic leakage in both *E. coli* (Fig. 4d) and *S. aureus* (Fig. 4e), and bacteria with H₂O₂+US maintained intact morphology. Besides, element mapping of bacteria showed intracellular copper content was distinct in PH-CpBT+H₂O₂+US compared with US+H₂O₂ and PH-CpBT+H₂O₂ (Supplementary Fig. 17b), indicating an increased intracellular copper inward flow due to the membrane permeability alteration induced by evaluated ROS (Fig. 4f and Supplementary Fig. 18). Previous reports showed that Cu ions have dose-dependent antibacterial activity²². To find out the antibacterial effect of Cu ions in this work, Cu ions

released from PH-CpBT scaffolds with or without US stimulation were analyzed by an ICP spectrometer (Supplementary Fig. 19). The total amount of released Cu ions was 3-7 $\mu\text{g/L}$, much lower than MIC of Cu^{2+} ions (630 $\mu\text{g/L}$ for *S. aureus* and 63-630 $\mu\text{g/L}$ for *E. coli*)³². Furthermore, *S. aureus* or *E. coli*. were co-incubated with the leaching solution of PH-CpBT after US stimulation (30 min) for 24 h (Supplementary Fig. 20a), finding that the antibacterial rate was lower than 50%, indicating that the antibacterial performance by only released Cu ions was poor. Besides, bacteria were incubated with PH-CpBT+US and tetrathiomolybdate (TTM, a copper chelator). In this situation, Cu ions cannot get into the bacteria, resulting in an antibacterial rate of 70% due to ROS attack (Supplementary Fig. 20b). Furthermore, mBT and Cu ions coating on Pp without pDA as interlayer exhibited a lower antibacterial rate than PH-CpBT (Supplementary Fig. 20c), suggesting that the electron transport within pDA was a key factor for the reduction of Cu^{2+} . These results proved that the antibacterial mechanism was more intricate than the release of Cu ions.”

Besides, we added additional animal experiments to clarify the role of Cu ions in vivo to the Revised Manuscript on page 19, line 24:

“Visual examination revealed the presence of secretions and pus at the implant site of Pp and PH, with partial mitigation observed in PH-pBT, PH-CpBT (US-), and PH-mBT/Cu (Supplementary Fig. 28). By comparison, the implant site of PH-CpBT+US and vancomycin (Van.) exhibited smooth tissue healing without secretion and pus formation, suggesting the clearance of bacterial infection. The order of bacterial colonies on agar plates and the turbidity of the Luria-Bertani medium after cultivation were as follows: Pp \approx PH > PH-pBT \approx PH-CpBT (US-) > PH-mBT/Cu > PH-CpBT \approx Van, suggesting that PH-CpBT exhibited favorable in vivo antibacterial characteristics rivaling Van (Fig. 5b and 5c). For PH-mBT/Cu, the lack of pDA as “electron aspirator” to initiate the reduction of Cu^+ , the antibacterial performance by SDT and released Cu^{2+} only was inefficient. For PH-pBT, no induction of copper-induced cuproptosis-like bacterial death occurred due to the absence of Cu ions, also exhibited moderate antibacterial effect.”

9. The antimicrobial performance or biofilm elimination ability of this nanoreactor CpBT should be compared with similar materials to demonstrate the advantages of this material.

Answer: Thank you for your comment. To compare the antibacterial performance of CpBT with similar materials, including commercial pure BaTiO₃ (Macklin Inc.), ZnO (Macklin Inc.), MoS₂, and TiO₂ (Sinopharm Chemical Reagent Co., Ltd.) were subjected to the antibacterial experiment with US stimulation (Figure R8). The first three were classic piezoelectric material (*Adv. Mater.* 2019, 31, 1802084), and the last was sonosensitizer that was often investigated in SDT (*J. Am. Chem. Soc.* 2020, 142, 6527–6537). The particle size of above materials was about 100 nm. It can be seen that although these groups showed certain antibacterial performance, with the antibacterial rate lower than 70%. Obviously, CpBT showed better antibacterial properties due to ultrasound-activated piezo-hot carriers triggering tandem catalysis coordinating cuproptosis-like bacterial death.

Figure R8 a Plate cultures of *S. aureus* and *E. coli* treated with different materials and US stimulation.

The related antibacterial rate against **b** *S. aureus* and **c** *E. coli*.

We have added more discussion to the Manuscript on page 15, line 1:

“In addition, we compared the antibacterial activity of CpBT with other piezoelectric materials (including BT, ZnO, MoS₂ and TiO₂), finding that CpBT showed better antibacterial activity (Supplementary Fig. 21).”

10. In the in vivo experiments, the authors did not record the weight of each group of mice statistically, so there is no guarantee that each group of mice is basically the same.

Answer: Thank you for your comment. Initially, rats weighing approximately 250 g were procured for our study. They were randomly caged and underwent a about two-week acclimatization period before the commencement of the experiments. Subsequently, during the experimental grouping, rats from the same cage were evenly distributed across various test groups to ensure baseline uniformity. Additionally, we measured weight before experiment, confirming no significant differences in the initial weights among groups (**Figure R9**).

Figure R9 The initial body weight of rats for different groups.

We have added more discussion to the Manuscript on page 19, line 19:

“The baseline weights of all groups were equivalent before surgery (Supplementary Fig. 27).”

11. It is suggested that the relevant content of the H&E images should be labelled, such as

lymphocytes, and the quantitative analysis should be added for better observation and understanding by readers.

Answer: Thank you for your comments and suggestions. The nuclei of neutrophils appeared dark purple, and the cytoplasm appeared pale pink. For better observation, the neutrophils and lymphocytes were marked with red and green arrows, respectively (Fig. 5d). And the related quantitative analysis was shown in Fig. 5e.

Fig. 5 d H&E images and **e** semi-quantification of neutrophil in the infected bone tissues surrounding the implants. The red arrows represented neutrophils, and the green arrows represented lymphocytes.

12. Statistical analysis should be added to the CT range area in Fig. 6a to visualize the speed of new bone growing.

Answer: Thank you for your comment. On one hand, we demonstrated the growth rate and overall quantity of new bone in PH-CpBT group at different stages through videos (Supplementary Information, Video 1), where the red portions represented newly formed bone. On the other hand, in Fig. 6a, we incorporated the amount of new bone at each stage, providing a clearer depiction of the new bone growth rates at 4 and 8 weeks.

Figure R10 Typical screenshots of bone generation video.

Fig. 6 The number and speed of new bone growing into the scaffold. **a** Micro-CT of femoral condyle, from up to down, reconstruction of defect and new bone ingrowth in the scaffolds, top view of new bone, side view of new bone. Quantitative statistics of bone regeneration related index in 3D reconstruction by micro-CT including.

13. DAPI is a DNA dye that clearly shows the nucleus and thus the distribution of cells, and should be the largest of all stains, while the range of DAPI in the stained picture in Fig. 7d is too small

Answer: Thank you for your reminding. In the previous version, DAPI color was overlooked to emphasize RUNX2, BMP-2, and CD31. Indeed, in the overall staining, DAPI should exhibit the strongest expression, even in Pp group. We have updated the images in Fig. 7d to rectify this, and the alterations do not affect the differences between the groups in terms of osteogenesis and angiogenesis.

Fig. 7 d Immunofluorescence staining of CD31, BMP-2, and RUNX2 surrounding tissues of im-plants at 8 weeks post-surgery.

14. There is no description of the statistical analysis in the figure notes to Figs. 4 and 7, and there is no specific p-value range for **** in Figs. 5 and 6.

Answer: We thank the reviewer for careful reading of our manuscript. We have added statistical descriptions in the notes to Figs. 4 and 7: “The experiment was repeated three times independently, with a representative example shown. Significant differences between groups are indicated as **** $p < 0.0001$, *** $p < 0.001$, ** $p < 0.01$, and * $p < 0.05$.”. At the same time, the meaning of **** was added in Figs. 5 and 6 “Significant differences between groups are indicated as **** $p < 0.0001$ ”.

15. Fig. S1, Fig. S12 and Fig. S26 have problematic figure notes.

Answer: Thank you for your comment. We have corrected it.

16. No scale bars have been added to Fig. S30.

Answer: We thank the reviewer for careful reading of our manuscript. We have added the scale bars matching 100 μm to Fig. S30.

Reviewer #2

Comments:

Huang et al. have developed a novel barium titanate-composite antibacterial coating for PEKK scaffolds. This coating employs a unique Tandem Catalysis mechanism, combining SDT and CDT to effectively combat bacterial infections. While their work is substantial, several significant shortcomings have led me to conclude that it may not be suitable for publication in Nature Communications. The specific issues are outlined below:

Answer: Thank you very much for your comments. We find your comments and suggestions are of benefit to improving our manuscript. We have made major modifications in the Revised Manuscript to address the role of Cu^{2+} in our systems, and do more experiments to clarify the real antibacterial mechanism of CpBT nanoreactors. Specific explanations are shown below.

1. The author mentioned that the Cu^{2+} and single SDT is not active. However, numerous studies have demonstrated the potential of SDT in both tumor and bacterial eradication. Additionally, Cu^{2+} ions are widely recognized for their antibacterial properties in tissue engineering. Therefore, the authors should provide a more compelling explanation in their manuscript, addressing my primary concern: whether the notable antibacterial effect arises from an enhanced release of Cu^{2+} ions induced by ultrasound.

Answer: We are grateful to the reviewer for their meticulous examination of our manuscript and their insightful comments. We are sorry that we haven't expressed the word "inactive" correctly. In this work, the word "inactive" in Fig. 1 meant to describe that the antibacterial performance of the single SDT by mBT or CDT by Cu^{2+} seemed to be not that effective based on our experimental results. Although preclinical studies on SDT have shown promising results in various cancer cells or bacteria, the therapeutic efficacy for eliminating tumors or biofilm-related infection in vivo is still unsatisfactory. Therefore, numerous novel nanoplatforms are applied in this emerging field to tackle these intrinsic barriers and achieve continuous innovations. In particular, the combination of SDT with other treatment strategies (such as PDT, CDT, immunotherapy, chemotherapy, gas therapy, and so on) has demonstrated superior efficacy in improving anticancer or antibacterial activity relative to that of monotherapies alone

(*Chem. Soc. Rev.* 2021, 50, 11227-11248; *Adv. Mater.* 2020, 32, 2003214; *Adv. Mater.* 2023, 35, 2211130). Besides, we admitted that antibacterial performance of Cu^{2+} ions is quite good, but it depends on the concentration, which was low in our work (please see below). To avoid misunderstanding, the word “inactive” in Fig. 1 was placed by “inefficient”.

Besides, we fully agreed that it was critical to demonstrate the antibacterial effect of the released Cu ions from PH-CpBT scaffold by ultrasound, and we thank the reviewer for pointing it out. A series of experiments were conducted to clarify this issue:

Firstly, Cu^{2+} ions released from PH-CpBT scaffolds at 1-, 4-, and 7-day intervals and with or without US stimulation (10 min/day, 1 MHz, 1.0 W/cm², 50% duty cycle, the same as in vivo treatment parameters) were analyzed by an inductively coupled plasma (ICP) spectrometer. Specifically, PH-CpBT scaffolds were immersed in a 0.9 % NaCl solution at 37 ± 1 °C with the surface-area to -volume ratio was 3 cm²/mL according to international standard ISO 10993-12. Triplicate samples were used to obtain an average value with standard deviation. As shown in **Figure R1a**, the total amount of released Cu^{2+} ions were 3.4, 4.0, and 4.3 $\mu\text{g/L}$ after 1, 4, and 7 days of immersion, respectively. The ultrasound stimulation will accelerate the release of Cu^{2+} ions, reaching 4.6, 6.3, and 7.3 $\mu\text{g/L}$ at 1, 4, and 7 days. It was reported that the antimicrobial minimum inhibitory concentration (MIC) of Cu^{2+} ions was 630 $\mu\text{g/L}$ for *S. aureus* and 63-630 $\mu\text{g/L}$ for *E. coli*. (*Chem. Res. Toxicol.* 2015, 28, 1815-1822), which was more than 10 times higher than the release concentration of Cu^{2+} in this work. Besides, for antibacterial Cu-contained metal or alloys, we surveyed the existing scientific literature and statistically integrated the findings, and the relationship between the antibacterial rate and Cu^{2+} ion release concentration was shown in **Figure R1b**. It showed that the low Cu^{2+} release concentration (<10 $\mu\text{g/L}$) resulted in bad antibacterial rate (<70%), and good antibacterial performance (>90%) required higher Cu^{2+} concentration (>100 $\mu\text{g/L}$). However, in this work, the low Cu^{2+} release concentration from PH-CpBT scaffolds (<8 $\mu\text{g/L}$ with US stimulation) resulted in good antibacterial rate (99.95%). Therefore, apart from Cu^{2+} release, US-activated piezo-hot carriers triggering tandem catalysis played an important part in antibacterial performance. Besides, the enrichment of Cu ions in bacteria was revealed by examining the ultrastructure of the bacteria using Bio-TEM (**Figure R1c**). Compared with US+H₂O₂ and PH-CpBT+H₂O₂ group,

intracellular copper content was distinct in PH-CpBT+H₂O₂+US, indicating intracellular copper inward flow due to the membrane permeability alteration induced by evaluated ROS. The antibacterial mechanism of Cu ions at low concentration in this work may be different from the previous reports.

Figure R1 a Cu²⁺ released from PH-CpBT scaffold in the 0.9 % NaCl solution at different immersion time with or without US stimulation. **b** Summary of antibacterial rates and Cu ions released concentrations of the reported antibacterial alloys (*Mat. Sci. & Eng. C* 2020, 115, 1; *Biomed. Mater.* 2014, 9, 025013; *Mat. Sci. & Eng. C* 2016, 69, 1210–1221; *Mat. Sci. & Eng. C* 2020, 115, 110921; *Mat.*

Sci. & Eng. C 2014, 35, 392–400; *Bioac. Mater.* 2020, 5, 659–666; *Surf. Coat. Tech.* 2021, 421, 127438; *Mater. Technol.* 2015, 30: 6, 68-72; *Chem. Res. Toxicol.* 2015, 28, 1815-1822). **c** Bio-TEM images of bacteria and Cu element mapping in different groups.

Secondly, to explore the antibacterial performance of released Cu^{2+} in this work, we did a series of antibacterial experiments (**Figure R2**). Firstly, *S. aureus* or *E. coli*. were co-incubated with the leaching solution of PH-CpBT scaffold after US stimulation (30 min) for 1 day (Figure R2a), finding that the antibacterial rate was lower than 50% (Figure R2e-g), indicating that the antibacterial performance by released Cu ions only was not that good.

Thirdly, to exclude the effect of released Cu ions on antibacterial effect, tetrathiomolybdate, a copper chelator [*Science* 2022, 375(6586): 1254-1261; *Dalton Trans.* 2022, 51(27): 10361-10376; *J. Neurol. Sci.* 1973, 20, 63-72], was co-incubated with bacteria and PH-CpBT (Figure R2b) under US stimulation. In this situation, Cu ions cannot get into the bacteria, so the antibacterial effect only came from ROS production by tandem catalysis between SDT and CDT, representing an antibacterial rate of ~70%, which indicated that Cu-related cuproptosis-like bacterial death was also important antibacterial mechanism in this work.

At last, in CpBT system, pDA served a crucial function as an electron transporter in the conversion of Cu ions. Therefore, Pp scaffold was coated with mBT and Cu ions, whose content was equal to CpBT on PH-CpBT scaffold (Figure R2c, defined at PH-mBT/Cu group). In this situation, the antibacterial effect originated from ROS generated by SDT of mBT, and Cu^{2+} release with US stimulation. The antibacterial rate of PH-mBT/Cu was ~75%, indicating that pDA, which acted as the “electron aspirator” to extract US-activated piezo-hot carriers and initiated the oxidizing reaction of Cu^{2+} to Cu^+ , played an important role in this work.

As a result, apart from Cu^{2+} release, US-activated piezo-hot carriers trigger tandem catalysis played an important part in antibacterial performance, and increased copper accumulation occurred only under the influence of ROS, contributing to cuproptosis-like bacterial death caused by copper.

Figure R2 a-d Schematic diagram for different antibacterial experiments. **e** Typical images of *S. aureus* and *E. coli* colonies treated by various groups. The corresponding antibacterial rate against **f** *S. aureus* and **g** *E. coli* after different treatments. The experiment was repeated three times independently, with a representative example shown. Significant differences between groups were indicated as **** $p < 0.0001$, *** $p < 0.001$, ** $p < 0.01$, and * $p < 0.05$.

To clarify the real antibacterial mechanism in this work, we have added more discussion to the Manuscript on page 4, line 20:

“However, Cu ions have dose-dependent antibacterial performance and cytotoxicity²², and whether Cu ions can induce cuproptosis-like bacterial death in a concentration lower than the minimal inhibitory concentration (MIC) and non-cytotoxic extracellular aggregation remains to be explored.”

Page 14, line 7:

“To observe microscopic changes in dead bacteria, Bio-TEM images of bacteria were examined (Supplementary Fig. 17a). Bacteria incubated with PH-CpBT+H₂O₂+US showed incomplete walls and cytoplasmic leakage in both *E. coli* (Fig. 4d) and *S. aureus* (Fig. 4e), and bacteria with H₂O₂+US maintained intact morphology. Besides, element mapping of bacteria showed intracellular copper content was distinct in PH-CpBT+H₂O₂+US compared with US+H₂O₂ and PH-CpBT+H₂O₂ (Supplementary Fig. 17b), indicating an increased intracellular copper inward

flow due to the membrane permeability alteration induced by evaluated ROS (Fig. 4f and Supplementary Fig. 18). Previous reports showed that Cu ions have dose-dependent antibacterial activity²². To find out the antibacterial effect of Cu ions in this work, Cu ions released from PH-CpBT scaffolds with or without US stimulation were analyzed by an ICP spectrometer (Supplementary Fig. 19). The total amount of released Cu ions was 3-7 $\mu\text{g/L}$, much lower than MIC of Cu^{2+} ions (630 $\mu\text{g/L}$ for *S. aureus* and 63-630 $\mu\text{g/L}$ for *E. coli*)³². Furthermore, *S. aureus* or *E. coli*. were co-incubated with the leaching solution of PH-CpBT after US stimulation (30 min) for 24 h (Supplementary Fig. 20a), finding that the antibacterial rate was lower than 50%, indicating that the antibacterial performance by only released Cu ions was poor. Besides, bacteria were incubated with PH-CpBT+US and tetrathiomolybdate (TTM, a copper chelator). In this situation, Cu ions cannot get into the bacteria, resulting in an antibacterial rate of 70% due to ROS attack (Supplementary Fig. 20b). Furthermore, mBT and Cu ions coating on Pp without pDA as interlayer exhibited a lower antibacterial rate than PH-CpBT (Supplementary Fig. 20c), suggesting that the electron transport within pDA was a key factor for the reduction of Cu^{2+} . These results proved that the antibacterial mechanism was more intricate than the release of Cu ions.”

2. Following above, they must test the ion valence state of Cu in this reaction or the solution.

The real anti-bacterial mechanism should be better revealed.

Answer: Thank you for your comment. To investigate the conversion of valence state of Cu ions during US stimulation, neocuproine, a Cu^+ -specific sequestering agent, was used as an indicator [*Eur. J Pharmacol.* 2009, 605(1-3): 158-163]. In the detection of Cu^+ , colorless neocuproine typically formed yellow or deep red complexes $[\text{Cu}(\text{neocuproine})_2]^+$ (**Figure R2a**), and the resulting yellow-colored chromophore showed maximum absorption at 452 nm. After US stimulation, the neocuproine solution treated with CpBT turned yellow, providing strong evidence of the formation of Cu^+ on the CpBT nanoreactor during US stimulation. Besides, the characteristic absorption peak of $[\text{Cu}(\text{neocuproine})_2]^+$ increased slowly with US time increased, suggesting that the production of Cu^+ was time-dependent.

Figure R3 **a** The reaction mechanism of Cu⁺ detection by neocuproine. **b** UV-Vis absorbance spectra of neocuproine solution treated with CpBT and US stimulation.

We have added more discussion to the Manuscript on page 8, line 10:

“To figure out enhanced ROS production in the presence of H₂O₂, neocuproine, a Cu⁺-specific sequestering agent, was used as an indicator. In the detection of Cu⁺, colorless neocuproine typically formed yellow complexes [Cu(neocuproine)₂]⁺ (Fig. 2h), showing maximum absorption at 452 nm. Neocuproine solution treated with CpBT+US turned yellow (Fig. 2h), providing strong evidence of the formation of Cu⁺ on CpBT nanoreactor during US stimulation. Besides, the characteristic absorption peak increased slowly with US time increased, suggesting that the production of Cu⁺ was time-dependent (Supplementary Fig. 6a). Without piezo-electrons produced by mBT, the reduction of Cu²⁺ to Cu⁺ under US stimulation failed (Supplementary Fig. 6b and c).”

3. In the process of preparing the coated scaffold, the authors solely employ the method of solution soaking. This prompts the question of whether the antibacterial effect primarily results from ion release in a water-based environment. The stability of the coating under such conditions warrants investigation.

Answer: We thank the reviewer for the careful reading of our manuscript. As we answered in Q1, the released concentration of Cu ions from PH-CpBT scaffold is quite low. PEKK possesses good surface chemical modification potential, which is easier to modify. In this work, PEKK scaffolds were modified with polydopamine (defined as Pp scaffolds), then Pp scaffolds were further modified with CpBT. Polydopamine (pDA) coating of surfaces was a versatile strategy

to fabricate functional films on various substrates. As a mussel-inspired material, polydopamine, which possessed many properties, such as a simple preparation process, good biocompatibility, strong adhesive property, and easy functionalization, can easily functionalize broader range of surfaces, including most noble metals and metal oxides as well as materials with low surface energy (titanium alloy, carbon nanotubes, graphene, PS, polyether-ether-ketone etc.) (*Nat. Commun.* 2023, 14, 664; *ACS Nano* 2019, 13, 8537–8565; *Ultrason. Sonochem.* 2021, 74, 105571). The decoration of PDA films on these materials not only improves their solubility and stability but also renders them multifunctional and smart platforms.

Furthermore, supposed that the pDA coating was not that stable, CpBT nanoreactor would fall off PH-CpBT scaffold, along with the released Cu ion from CpBT, as shown in Figure R4 (process A). We collected the leaching solution of PH-CpBT after US (10 mins/day) for 7 days, and detected the concentration of Cu ions by ICP was ~ 8 $\mu\text{g/L}$. In process B, we used the centrifugal machine to exclude CpBT nanoreactor in the leaching solution, and the concentration of Cu ions detected by ICP was ~ 7 $\mu\text{g/L}$. The concentration of Cu ions in two processes was similar, indicating that the content of CpBT nanoreactor in leaching solution was low. The results suggested that the pDA coating was stable with US stimulation.

Figure R4 Cu ions released from PH-CpBT scaffold with different treatments.

We have added more discussion to the Manuscript in page 13, line 20:

“To assess the stability of CpBT coating on PH-CpBT with US stimulation, the released CpBT

was detected by inductively coupled plasma (ICP) (Supplementary Fig. 14). Process A contained the released Cu ions and CpBT from scaffolds, and process B contained the released Cu ions only. It showed that the concentration of Cu element of two processes by ICP was similar, indicating that CpBT on PH-CpBT surface was stable due to the strong adhesion of pDA³¹.”

4. Furthermore, the authors have only presented the antibacterial properties of the powder form of their material. It is essential to assess the catalytic ability of the scaffold itself, as this will provide a more comprehensive understanding of its antibacterial potential.

Answer: We thank the reviewer for careful reading of our manuscript. All the antibacterial experiments in this work were carried out with scaffolds. Taking the antibacterial plate experiment as an example, following the scaffold preparation method outlined earlier, we obtained three scaffolds for each group. Subsequently, the scaffolds were placed into bacterial suspensions at a scaffold surface area/volume ratio of 1.5. After US stimulation for 10 mins, an appropriate bacterial suspension was taken for plate observation.

5. Additionally, some studies also reported the potential for ultrasound to facilitate the conversion of Cu²⁺ ions into Cu⁺ ions. Consequently, it is imperative to reconsider and conduct further experiments to explore the relevance of SDT and CDT in this work. How important role of SDT play in this process?

Answer: Thank you for your comment. To our knowledge, the conversion of Cu²⁺ to Cu⁺ in vivo was mostly realized by glutathione (enrichment in tumor tissue) (*J. Am. Chem. Soc.* 2019, 141, 849-857; *J. Am. Chem. Soc.* 2023, 145, 4279-4293), reducing agent [such as sodium ascorbate, *J. Am. Chem. Soc.* 2023, 145(3): 1955-1963], the localized surface plasma resonance effect triggered by light (*Biomaterials* 2020, 255, 120167; *Nano Today* 2022, 43, 101397), and X-ray (*Nano Lett.* 2019, 19, 1749-1757). There were few reports on *in situ* conversion of Cu²⁺ to Cu⁺ triggered by low-intensity ultrasound that applied in this work. Pedro *et al.* reported that ultrasound-assisted Cu⁺-catalyzed alkyne-azide click reaction by metallic copper [*Nat. Protoc.* 2010, 5(3), 607], and the ultrasound used in this work was high-power US probe system (21

kHz, 25 W) working at higher temperature of 70 °C or 100 °C. Another work by Chen *et. al.* reported that ultrasound-driven bioorthogonal catalytic therapy through ultrasmall poly(acrylic acid)-modified copper nanocomplexes (Cu@PAA NCs) (*Adv. Mater.* 2023, 35, 2209179). They proposed that the electrons can be ejected from the surface of Cu@PAA NCs through ***the photoelectric effect by US-mediated sonoluminescence***, which converted Cu²⁺ to Cu⁺. The conversion mechanism that related to photo-electrons was different from our work that the conversion of Cu²⁺ to Cu⁺ in our work was triggered by US-induced piezo-hot carriers.

Besides, to figure out whether ultrasound can facilitate the conversion of Cu²⁺ into Cu⁺ ions in this work, we synthesized pDA nanoparticles, and the same amount of Cu²⁺ was chelated to the surface of pDA (pDA@Cu). Then, we used the specific reagent neocuproine to detect whether the valence state of Cu²⁺ on pDA@Cu will change under US. The result was shown in Figure R5a, the characteristic absorption peak of the [Cu(neocuproine)₂]⁺ did not increase as time went by, suggesting that the conversion of Cu²⁺ to Cu⁺ failed in absence of piezoelectric mBT. Besides, we mixed free Cu²⁺ (from CuSO₄) with neocuproine, and the color of solution also did not change, indicating that low-intensity ultrasound (1.0 W/cm², 50% duty, 1 MHz) cannot change the valence state of Cu²⁺ (Figure R5B). It can draw a conclusion that piezoelectricity of mBT that showed SDT performance played a crucial role in this process.

Figure R5 **a** UV-Vis absorbance spectra of pDA@Cu and **b** Cu²⁺ ions with US stimulation.

Reviewer #3

Comments:

Huang et al. present a report on the design and anti-bacterial function of US-activated piezohot carriers. The study revealed that the carrier could induce the valence state interconversion between Cu^{2+} and Cu^+ , and amplify the ROS generation via Cu^+ -catalyzed chemodynamic reactions or copper overload-induced interruption of the tricarboxylic acid cycle. The manuscript is well-written and contains substantial data. The authors mentioned that metabolic programming played a role in the anti-bacterial function of the carrier, however, the analysis of data from metabolomics was missing. I think it is necessary to include the data from metabolomics in the paper or supplementary information, as suggested below.

Answer: Great thanks to you for your earnest work and affirmation to our work. We have revised our manuscript according to your questions carefully. Specific explanations are shown below.

1. The author should clearly describe the differential metabolites in Control, PH-CpBT+ H_2O_2 , and PH-CpBT+US, as they did in Fig. 4K.

Answer: We thank the reviewer for careful reading of our manuscript and his/her insightful comment. As the reviewer pointed out, to precisely reflect the impact of CpBT on TCA cycle-related metabolites, we performed targeted determinations of carbon cycle metabolites for three groups: Control, PH-CpBT+ H_2O_2 , and PH-CpBT+ H_2O_2 +US. This approach was chosen to mitigate the uncertainties associated with untargeted metabolite assessments. Notably, within the overall metabolism and TCA cycle pathways, PH-CpBT+ H_2O_2 +US group exhibited lower metabolite expression compared with Control and PH-CpBT+ H_2O_2 (Figure R 1), indicating the inhibition of the overall TCA cycle by PH-CpBT+ H_2O_2 +US. This aligned with the function of copper-induced cell death through TCA cycle enzyme binding. It was noteworthy that PH-CpBT+ H_2O_2 group displayed an overall higher metabolite profile (Figure R1b), suggesting that increased copper intracellular transport may stimulate overall bacterial metabolism and energy requirements. Furthermore, to validate the changes in the depicted metabolites in Fig. 4k, we conducted experiments on bacterial membrane disruption, DNA, and protein leakage under

oxidative stress. The PH-CpBT+H₂O₂+US group exhibited more DNA and protein leakage, indicating greater bacterial membrane disruption (Figure R2).

Figure R1 Differential heat map of TCA cycle metabolites. **a** PH-CpBT+H₂O₂+US vs. Control. **b** PH-CpBT+H₂O₂+US vs. PH-CpBT+H₂O₂.

Figure R2 Quantitative analysis of **a** protein leakage and **b** the intact bacterial DNA of *S. aureus* on different scaffolds after different treatments.

- It would be preferable if the author could perform a correlation analysis between differential metabolites and differentially expressed genes, the analysis will give the reader a comprehensive view of signal transduction and regulation, gene expression, and dynamic metabolite changes.

Answer: Thank you for your comment. Understanding comprehensively the impact of PH-

CpBT on bacteria from a metabolic and genetic perspective is crucial. To depict this, we have clearly illustrated genes and metabolite processes related to the TCA pathway in the KEGG TCA pathway. It was evident that in PH-CpBT+H₂O₂+US group, more copper entered bacterial cells due to the destruction of bacterial membrane by ROS, a key factor in inhibiting the TCA cycle and inducing cuproptosis-like death. As emphasized in previous articles (*Science* 2022, 375(6586): 1254-1261), the binding of copper with acylated proteins led to their inactivation. In the TCA cycle, lipoylation of enzymes occurred at positions of DBT (Dihydrolipoamide Branched Chain Transacylase E2), GCSH (Glycine Cleavage System Protein H), DLST (Dihydrolipoamide S-Succinyltransferase), and DLAT (Dihydrolipoamide S-Acetyltransferase) [*J Biol. Chem.* 2018, 293, 7522-7530; *Science* 2022, 375(6586): 1254-1261]. This lipoylation, closely linked to the action of lipoic acid, responsible for enzyme acylation (*Curr. Opin. Chem. Biol.* 2018, 42, 76-85), was pivotal. Increased lipoylation led to decreased lipoic acid, and the reduction of lipoic acid negatively regulated the upregulation of genes such as lipI and lipA [*Microbiol. Mol. Biol. Rev.* 2016, 80(2):429-50]. An increased binding of Cu with lipoylation enzymes resulted in a reduction of metabolites in the TCA cycle, except for the upstream pyruvate (Figure R3). The decreased metabolites negatively regulated the bacterial system, leading to an upregulation of more TCA cycle genes. This also demonstrated that copper does not directly affect genes but first influences lipoylation proteins, thereby impacting the entire bacterial metabolic pathway.

Figure R3 A correlation analysis between differential metabolites and differentially expressed genes. Red rectangles indicated increased metabolites, and green rectangles indicate decreased metabolites; Red circles indicated up-regulated genes, and green ovals indicated down-regulated genes. DBT: Dihydrolipoamide Branched Chain Transacylase E2, GCSH: Glycine Cleavage System Protein H, DLST: Dihydrolipoamide S-Succinyltransferase, DLAT: Dihydrolipoamide S-Acetyltransferase.

We have added more discussion to the Manuscript on page 16, line 27:

“To present the changes in metabolites and the relevant genes in the TCA cycle more clearly, we conducted targeted detection of TCA cycle metabolites. We observed the lowest levels of metabolites in PH-CpBT+H₂O₂+US compared with Control and PH-CpBT+H₂O₂ (Supplementary Fig. S25). Fig. 4l vividly illustrated the changes in TCA cycle metabolites and the genes influenced by them. In short, under the influence of ROS, bacteria with PH-CpBT+H₂O₂+US experienced an increase in intracellular Cu, which bound with four lipoylation enzymes (DLAT, GCSH, DBT, and DLST)^{19,34}, leading to their deactivation and consequent reduction in metabolites throughout the pathway. Additionally, due to the formation of more lipoylation proteins, lipoic acid was reduced³⁵, resulting in the upregulation of the genes that regulated lipoic acid synthesis³⁶. Ultimately, the decreased metabolites, through negative feedback regulation, increased the expression of genes regulating the TCA cycle. Therefore,

through metabolomics and transcriptomics, it was evident that copper-induced cuproptosis-like death, causing an overall reduction in metabolism, and in synergy with ROS, effectively killed bacteria.”

With these modifications, we hope now the manuscript is acceptable for publication in Nature Communications.

Yours Sincerely,

Prof. Zongke Zhou,

Director, Department of Orthopaedics, Orthopaedic Research Institute

West China Hospital, Sichuan University, China

Email: zongke@126.com zhouzongke@scu.edu.cn

Prof. Xianzeng Zhang

Dean, College of Photonic and Electronic Engineering

Fujian Normal University, China

Email: xzzhang@fjnu.edu.cn

REVIEWERS' COMMENTS

Reviewer #1 (Remarks to the Author):

In my opinion, the revised manuscript answers all the issues raised from reviewers. The manuscript is now suitable for publication in Nature Communication.